# Inferring spikes from calcium imaging in dopamine neurons

**Weston Fleming**[1][ö], **Sean Jewell**[2][ö], **Ben Engelhard**[1], **Daniela M. Witten**[2]*, **Ilana B. Witten**[1]*

**1** Princeton Neuroscience Institute, Princeton University, Princeton, New Jersey, United States of America, **2** Department of Statistics & Biostatistics, University of Washington, Seattle, Washington, United States of America

ö These authors contributed equally to this work.

* iwitten@princeton.edu (IBW); dwitten@uw.edu (DMW)

**Data Availability Statement:** The dataset used in this study are available on figshare at DOI: https://doi.org/10.6084/m9.figshare.14627238.v1 Also, the software for performing spike inference is still

## Abstract

Calcium imaging has led to discoveries about neural correlates of behavior in subcortical neurons, including dopamine (DA) neurons. However, spike inference methods have not been tested in most populations of subcortical neurons. To address this gap, we simultaneously performed calcium imaging and electrophysiology in DA neurons in brain slices and applied a recently developed spike inference algorithm to the GCaMP fluorescence. This revealed that individual spikes can be inferred accurately in this population. Next, we inferred spikes *in vivo* from calcium imaging from these neurons during Pavlovian conditioning, as well as during navigation in virtual reality. In both cases, we quantitatively recapitulated previous *in vivo* electrophysiological observations. Our work provides a validated approach to infer spikes from calcium imaging in DA neurons and implies that aspects of both tonic and phasic spike patterns can be recovered.

## Introduction

The use of genetically encoded calcium indicators like GCaMP has made it possible to record neural activity from large populations of neurons with cell-type-specificity [1–3]. These advances have spurred development of spike inference methods to estimate the underlying spikes from the imaged calcium [4–14], but to date their application has been largely limited to cortical and hippocampal neurons [15–21]. Thus, it is unclear how well these algorithms perform on other cell types, particularly those with tonic levels of baseline activity. One population of particular interest is midbrain dopamine (DA) neurons, given the shift in recent years to record from these neurons through calcium imaging [22–27], combined with the lack of established and validated methods to relate calcium and electrophysiology data in these neurons.

Two challenges arise when trying to infer spike rates from a new cell type. The first is the acquisition of ground-truth data (simultaneous imaging and electrophysiology data from the same neurons), which is needed to validate a spike inference method in a new cell type. The second is that spike inference algorithms rely on tuning parameters, and it is unclear how to select them, especially on a new cell type. For example, DA neurons have distinct calcium

available on github at: https://github.com/
jewellsean/spike_tools.

**Funding:** Funding was from NYSCF (IBW); NSF
GRFP (WF); ARO W911NF1710554 (IW); NIH
grants R01 EB026908 (DMW), R01 DA047869
(DMW & IBW), U19 NS104648-01 (IBW); NIH
K99MH122657 (BE); and a Simons Investigator
Award in Mathematical Modeling of Living Systems
(DMW). IBW is a New York Stem Cell Foundation
—Robertson Investigator. The funders had no role
in study design, data collection and analysis,
decision to publish, or preparation of the
manuscript.

**Competing interests:** The authors have declared
that no competing interests exist.

currents [28–30] and buffering properties [31–33] compared to cortical neurons, which may affect the relationship between calcium fluorescence and spikes.

Because of these challenges, most imaging studies focused on subcortical neurons such as DA neurons have directly correlated calcium fluorescence to behavioral variables without accounting for the slow calcium dynamics [22–27, 34, 35]. This approach risks introducing inaccuracies in characterizing the relationship between neural activity and behavior and makes it difficult to relate findings from calcium imaging to those arising from electrophysiology.

Thus, to first address the lack of ground-truth data in the case of midbrain DA neurons, we collected simultaneous GCaMP imaging and cell-attached electrophysiology recordings *in vitro* (given the technical limitation of simultaneous *in vivo* measurements of these two signals). We then used these data to validate a recently-developed spike inference algorithm in these neurons [4].

Next, to address the issue of how to select the tuning parameters on new datasets when ground-truth spikes are not available (which is inevitably the case for subcortical *in vivo* imaging data), we made use of the fact that this spike inference algorithm depends on only two tuning parameters, both of which have a biophysical interpretation. The first corresponds to the decay rate of the calcium indicator, which we measured in our *in vitro* data. The second determines the average estimated firing rate of the recorded neurons, which we selected based on mean firing rates previously observed in midbrain DA neurons *in vivo* [36–39], and which can be measured by phototagging a new cell type of interest.

Using this straightforward approach to select the tuning parameters, we were able to quantitatively recapitulate multiple published properties of DA spiking from *in vivo* imaging data. During delivery of unexpected reward, inferred firing rates are of the appropriate magnitude and duration when compared to *in vivo* electrophysiology data. During omission of expected reward, pauses in firing in response to reward omission can be detected. Underscoring the utility of spike inference, inferred spikes, but not the calcium fluorescence itself, recapitulates the previously reported relationship between reward delivery and reward omission. In addition to these fast response properties, slow upward and downward ramps in inferred tonic firing rate as animals approach rewards can also be detected. On the other hand, inference does not capture certain features of previously published data, including the reported variability in tonic firing rates across DA neurons. Thus, we describe strengths as well as limitations of a simple approach for inferring spikes in DA neurons. We expect this approach to generalize to other subpopulations of genetically identified subcortical neurons.

## Results

### Validation of spike inference using simultaneous calcium imaging and electrophysiology *in vitro*

To determine the efficacy of spike inference in ventral tegmental area (VTA) DA neurons, we generated a ground-truth dataset by performing simultaneous calcium imaging and cell-attached electrophysiological recordings of spontaneously active, burst-firing DA neurons expressing either GCaMP6f or GCaMP6m (Fig 1A, S1 and S2 Figs; GCaMP6f at 30˚C, n = 12 recordings from 9 total cells; GCaMP6f at 37˚C, n = 24 recordings from 12 total cells; GCaMP6m at 37˚C, n = 19 recordings from 9 total cells). We performed these simultaneous measurements *in vitro*, given that combining these approaches *in vivo* in the VTA, which is located deep in the brain, is not currently feasible. In addition to the data presented in Fig 1, we also recorded from pacemaker cells which had faster GCaMP decay kinetics (S1 Fig).

To estimate spike times on the basis of calcium fluorescence, and to compare to the corresponding ground-truth spike measurements, we leveraged a generative model of the calcium

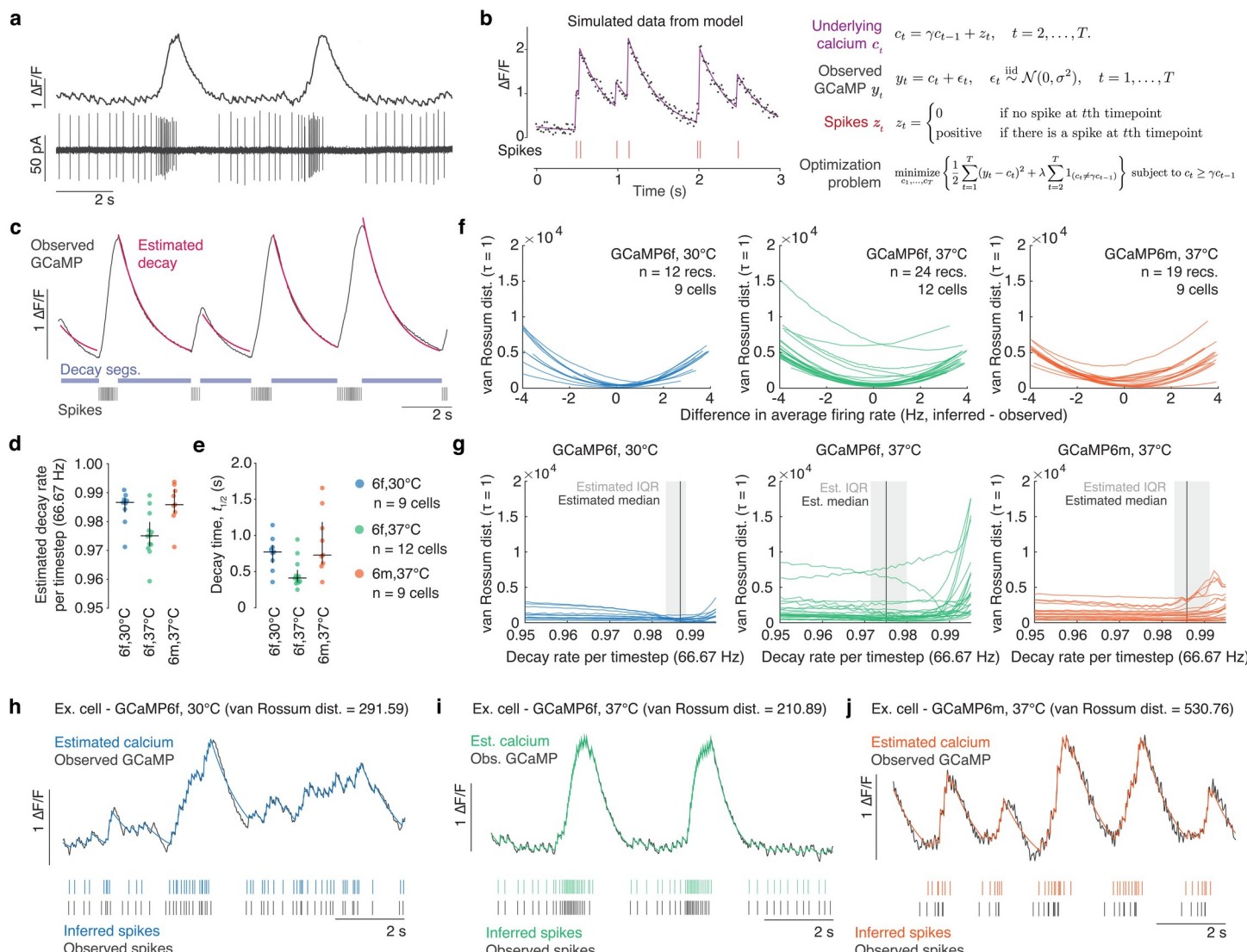

**Fig 1. Spike inference applied to VTA DA neurons *in vitro*.** a. Example data trace of simultaneous GCaMP6f (top) and raw cell-attached electrophysiology trace (bottom). b. Generative model of calcium dynamics. Simulated data (left) from this model (right) shows the underlying calcium $c_t$ (purple line) decays at rate γ until at time $s$ there is a spike, so that $z_s > 0$ (red vertical lines below). GCaMP observations $y_t$ (black dots) are noisy realizations of the underlying calcium concentration. c. Example of decay estimation using multiple spike-free segments ("Decay segs.", light blue). Observed GCaMP (black) decays at the same rate between spikes (black vertical lines below). The rate of exponential decay, γ, was estimated by fitting an exponential decay model to the observed GCaMP in the spike-free segments. d. Estimated decay rate per timestep at 66.67 Hz for GCaMP6f at 30˚C (median = 0.987, Q1 = 0.983, Q3 = 0.988), GCaMP6f at 37˚C (median = 0.975, Q1 = 0.971, Q3 = 0.980), and GCaMP6m at 37˚C (median = 0.986, Q1 = 0.983, Q3 = 0.991). e. Decay time (half life, $t_{1/2}$) for GCaMP6f at 30˚C (median = 0.772 s, Q1 = 0.619 s, Q3 = 0.866 s), GCaMP6f at 37˚C (median = 0.412 s, Q1 = 0.359 s, Q3 = 0.523 s), and GCaMP6m at 37˚C (median = 0.725 s, Q1 = 0.614 s, Q3 = 1.100 s). f. van Rossum distance for each experimental condition and for each recording, using the median decay rate across all recordings for that type of GCaMP and temperature, as a function of the difference between the average inferred firing rate and average observed firing rate. The distance is minimized when the average firing rate between the inferred and observed spikes is similar. g. van Rossum distance for each experimental condition and for each recording as a function of the decay rate γ when the tuning parameter is selected so that the inferred firing rate matches the observed firing rate of the recording. The shaded region and vertical dark grey line represent the estimated interquartile ranges and median values of the decay rate in d. h. Example of correspondence between observed GCaMP (black line, top) and estimated calcium (blue line, top), and observed spikes (black vertical dashes, bottom) and inferred spikes (blue vertical dashes, bottom) in a single cell expressing GCaMP6f measured at 30˚C. i. Example of correspondence between observed GCaMP (black line, top) and estimated calcium (green line, top), and observed spikes (black vertical dashes, bottom) and inferred spikes (green vertical dashes, bottom) in a single cell expressing GCaMP6f measured at 37˚C. j. Example of correspondence between observed GCaMP (black line, top) and estimated calcium (orange line, top), and observed spikes (black vertical dashes, bottom) and inferred spikes (orange vertical dashes, bottom) in a single cell expressing GCaMP6m measured at 37˚C.

dynamics resulting from spikes [8, 14] (Fig 1B). In this model, the fluorescence trace $y_t$ at the $t$th time step is a noisy observation of the underlying calcium concentration $c_t$, which is assumed to decay with rate $0<\gamma<1$ unless there is a spike. A spike causes an instantaneous increase in the calcium concentration (Fig 1B). As previously discussed [5], we fit the model by minimizing the difference between the observed fluorescence and modeled calcium, while applying an L0 regularization penalty on the total number of inferred spikes. A recently-developed dynamic programming approach is used to efficiently solve this non-convex optimization problem (Fig 1B; also see Methods).

This spike inference algorithm depends on only two parameters, both of which are easily interpretable: i) the calcium decay rate γ, and ii) the regularization tuning parameter λ, which controls the total number of inferred spikes in a recording—in other words, the average inferred firing rate. The decay rate γ was selected based on fitting an exponential curve to the GCaMP fluorescence in segments without spikes (Fig 1C–1E; see Methods for curve fitting).

To assess the agreement between the inferred spikes and the true spikes, we calculated the van Rossum distance metric for the inferred and true spikes for each neuron across a range of values of the tuning parameter λ (Fig 1F; see Methods) [40, 41]. This distance was smallest when the inferred average firing rate (computed as the number of inferred spikes divided by the length of the recording) equaled the true average firing rate, indicating that the true firing rate provides the optimal choice of λ. Based on this observation, to select λ for each neuron, we matched the inferred firing rate to the true firing rate of each neuron for our *in vitro* data.

To determine the sensitivity of spike inference to the decay rate γ, we calculated the van Rossum distance metric for the inferred and true spikes for each recording across a range of values of the decay rate, while matching the tuning parameter λ to the observed firing rate (Fig 1G). For most recordings, van Rossum distances were similar across the range of decay rates, particularly within the measured interquartile range of decay rates for each condition (light grey shaded regions in Fig 1G correspond to ranges in Fig 1D). This suggests that our spike inference will generate similar results even when the decay rate parameter does not exactly match the observed decay rate for that neuron. Based on this observation, we use the median decay rate of the respective GCaMP type for subsequent *in vitro* (Fig 1H–1J; S3–S5 Figs) and *in vivo* (Figs 2–4) estimations.

Across the population, inferred spike times tended to match true spike times in the *in vitro* dataset, during periods of tonic as well as burst activity (Fig 1H–1J, S3–S6 Figs). However, some patterns of activity were not well-recovered using this approach (S3–S6 Figs). There were occasional inaccuracies when estimating timing of the final spikes of a burst, or the first tonic spikes following a burst, when the GCaMP signal is still decaying (as seen in Fig 1J). Additionally, inferred spikes cannot recreate the timing of recorded spikes at frequencies above double the sampling frequency (66.67 Hz; see Methods).

In summary, our spike inference method appears well-suited to estimate both tonic and phasic activity in DA neurons, with some limitations on estimating exact spike placement during and immediately after burst activity, and with a limit to the maximum inferred firing rate. Given the successes and limitations in this *in vitro* data, we next asked whether our spike inference method could reproduce previously reported patterns of phasic and tonic activity in identified DA neurons *in vivo*.

## Application of spike inference to *in vivo* imaging data during the presentation of unexpected reward

While unexpected reward is known to generate widespread phasic responses in DA neurons based on electrophysiology data [39, 42–45], the extent to which spike inference can

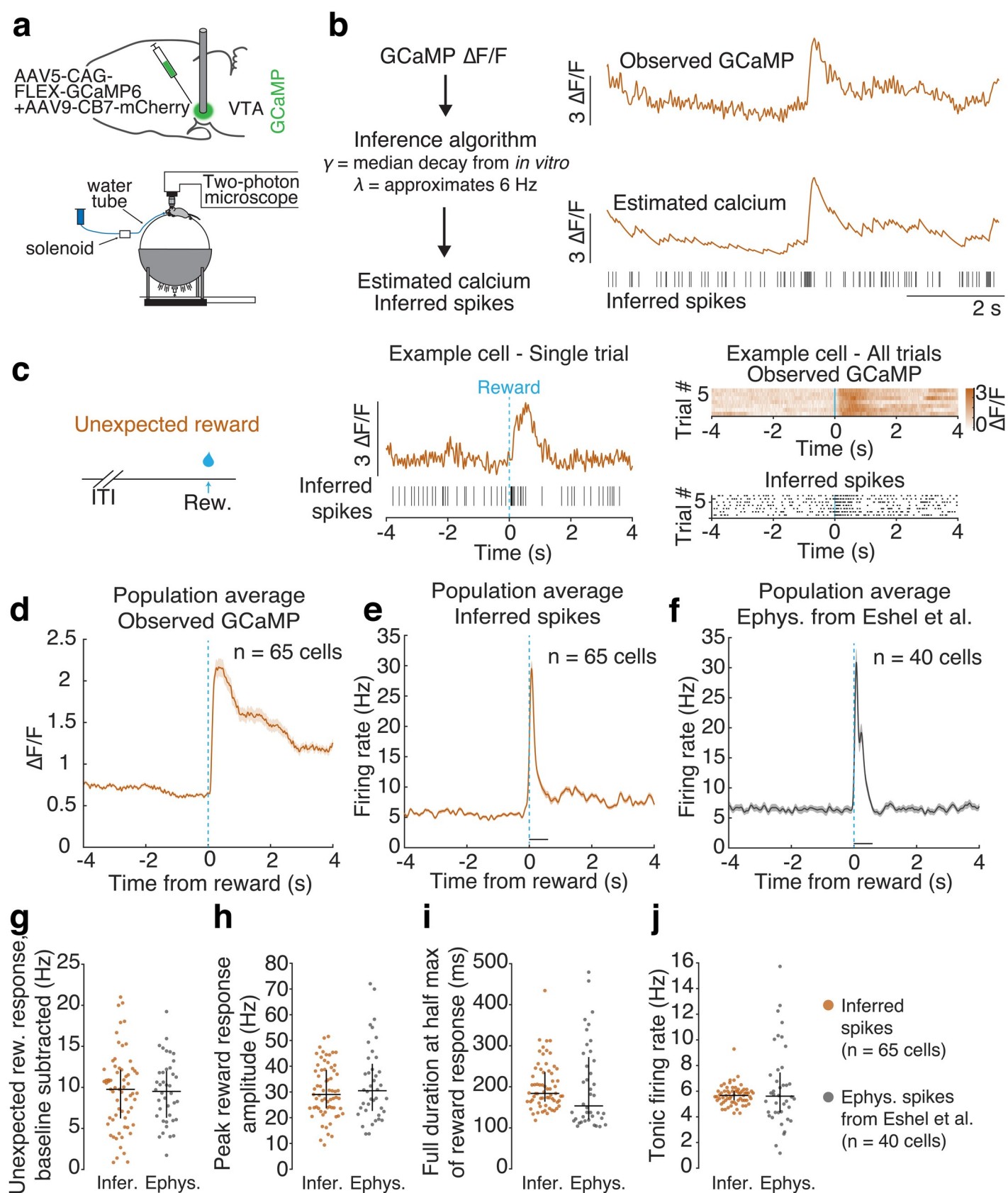

**Fig 2. Spike inference applied to *in vivo* calcium imaging in VTA DA during presentation of unexpected reward generates firing rate modulations comparable to those previously reported in other animals via electrophysiology.** a. Top: Schematic of the surgical strategy, where GCaMP6 is expressed in VTA DA neurons and a GRIN lens is implanted for imaging. Bottom: Schematic of recording setup, where the mouse is headfixed and VTA DA neurons are recorded via 2-photon calcium imaging. b. Schematic of spike estimation approach as applied to the *in vivo* data, where the observed ΔF/F is used to generate estimated calcium and inferred spikes using the decay parameter γ from *in vitro* experiment and a λ selected to target a 6 Hz average estimated firing rate. c. Left: Schematic showing unexpected reward is delivered after random inter-trial intervals. Middle: Example of an unexpected reward trial from a single cell, showing observed GCaMP (orange) and inferred spikes (black vertical lines). Right: All unexpected reward trials from an example cell, showing observed GCaMP (top) and inferred spikes (bottom). d. Mean observed GCaMP from population around presentation of unexpected reward (n = 65 cells). e. Mean population firing rate from inferred spikes from population around presentation of unexpected reward (n = 65 cells). f. Mean population firing rate from spikes recorded via electrophysiology (from Eshel et al. [39]; n = 40 cells) around presentation of unexpected reward. g-j. Comparison of inferred and electrophysiology spikes from Eshel et al. [39]. g. Unexpected reward response, where reward response is the mean firing rate over the first 600 ms following reward presentation, baseline subtracted using the mean firing rate over a 1 s period before reward presentation (inferred spikes over baseline median = 9.8 Hz, Q1 = 6.2 Hz, Q3 = 12.2 Hz; electrophysiology spikes over baseline median = 9.5 Hz, Q1 = 6.3 Hz, Q3 = 12.0 Hz). h. Peak reward response amplitude in inferred and electrophysiology spikes, where peak is maximum value of PSTH in the first 600 ms period following reward presentation (inferred spikes median = 29.0 Hz, Q1 = 23.7 Hz, Q3 = 38.6 Hz; electrophysiology median = 30.5 Hz, Q1 = 22.6 Hz, Q3 = 40.9 Hz). i. Full duration at half max of reward response peak in inferred and electrophysiology spikes (inferred spikes median full duration at half max = 183.9 ms, Q1 = 163.6 ms, Q3 = 235.9 ms; electrophysiology median full duration at half max = 153.5 ms, Q1 = 120.4 ms, Q3 = 272.5 ms). j. Mean tonic firing rates prior to presentations of unexpected reward, where tonic firing rates are calculated as the mean firing rate over a 1 s period prior to reward presentation (inferred spikes tonic firing rate median = 5.7 Hz, Q1 = 5.3 Hz, Q3 = 6.0 Hz; electrophysiology tonic firing rate median = 5.6 Hz, Q1 = 4.3 Hz, Q3 = 7.5 Hz). Vertical bars are interquartile range (Q1 and Q3). All data is from cells expressing GCaMP6f.

quantitatively recapitulate the magnitude and duration of the unexpected reward response has not been determined. Thus, we applied this spike inference algorithm to recently published *in vivo* two-photon calcium imaging data from VTA DA neurons during unexpected reward delivery [22] (Fig 2A–2E), and compared to the firing rates obtained from *in vivo* electrophysiology recordings from VTA DA neurons in a previously published dataset from other mice performing a different conditioning task [39]. For these spike estimates, the decay γ was set to the median decay rate from the *in vitro* measurements at 37˚C (γ = 0.970 for GCaMP6f 37˚C; γ = 0.984 for GCaMP6m at 37˚C at 60 Hz sampling frequency; Fig 1D), whereas the tuning parameter λ was set to correspond to a mean firing rate of 6 Hz, consistent with previous *in vivo* measurements of average DA neuron firing rates [36, 38, 39, 46].

We first compared the magnitude of the phasic response to unexpected reward in inferred spikes with spikes previously measured via electrophysiology in DA neurons in another group of mice (electrophysiology data from [39]). The unexpected reward response was defined as the baseline subtracted mean firing rate over a 600 ms window following reward onset (baseline calculated over a 1s period before reward) [39]. We found that inferred spikes showed a similar increase in firing rate following reward compared to the Eshel et al. electrophysiology data (inferred spikes median = 9.8 Hz, Q1 = 6.2 Hz, Q3 = 12.2 Hz; electrophysiology spikes median = 9.5 Hz, Q1 = 6.3 Hz, Q3 = 12.0 Hz; Fig 2E–2G). The peak amplitude of the unexpected reward response based on inferred spikes was also similar to those in the electrophysiology dataset (inferred spikes median = 29.0 Hz, Q1 = 23.7 Hz, Q3 = 38.6 Hz; electrophysiology median = 30.5 Hz, Q1 = 22.6 Hz, Q3 = 40.9 Hz; Fig 2H).

In contrast to the prolonged ΔF/F modulation to unexpected reward (Fig 2C and 2D), inferred spikes displayed a subsecond, phasic modulation (Fig 2C and 2E). In fact, the duration of the unexpected reward response was similar between inferred spikes and electrophysiology spikes (inferred spikes median full duration at half max = 183.9 ms, Q1 = 120.4 ms, Q3 = 272.5 ms; electrophysiology median full duration at half max = 153.5 ms, Q1 = 120.4 ms, Q3 = 272.5 ms; Fig 2I).

However, the electrophysiology population average firing rate displayed a biphasic response which was not evident in the inferred spikes (Fig 2F). It is unclear if this difference reflects a limitation of spike inference to detect the time course of a biphasic burst or is instead due to differences in the experimental design between the calcium imaging and the electrophysiology datasets. In the calcium imaging experiment, reward size was constant (reward size 8 μL), whereas for the electrophysiology dataset, reward size varied (ranging from 0.2 to 20 μL).

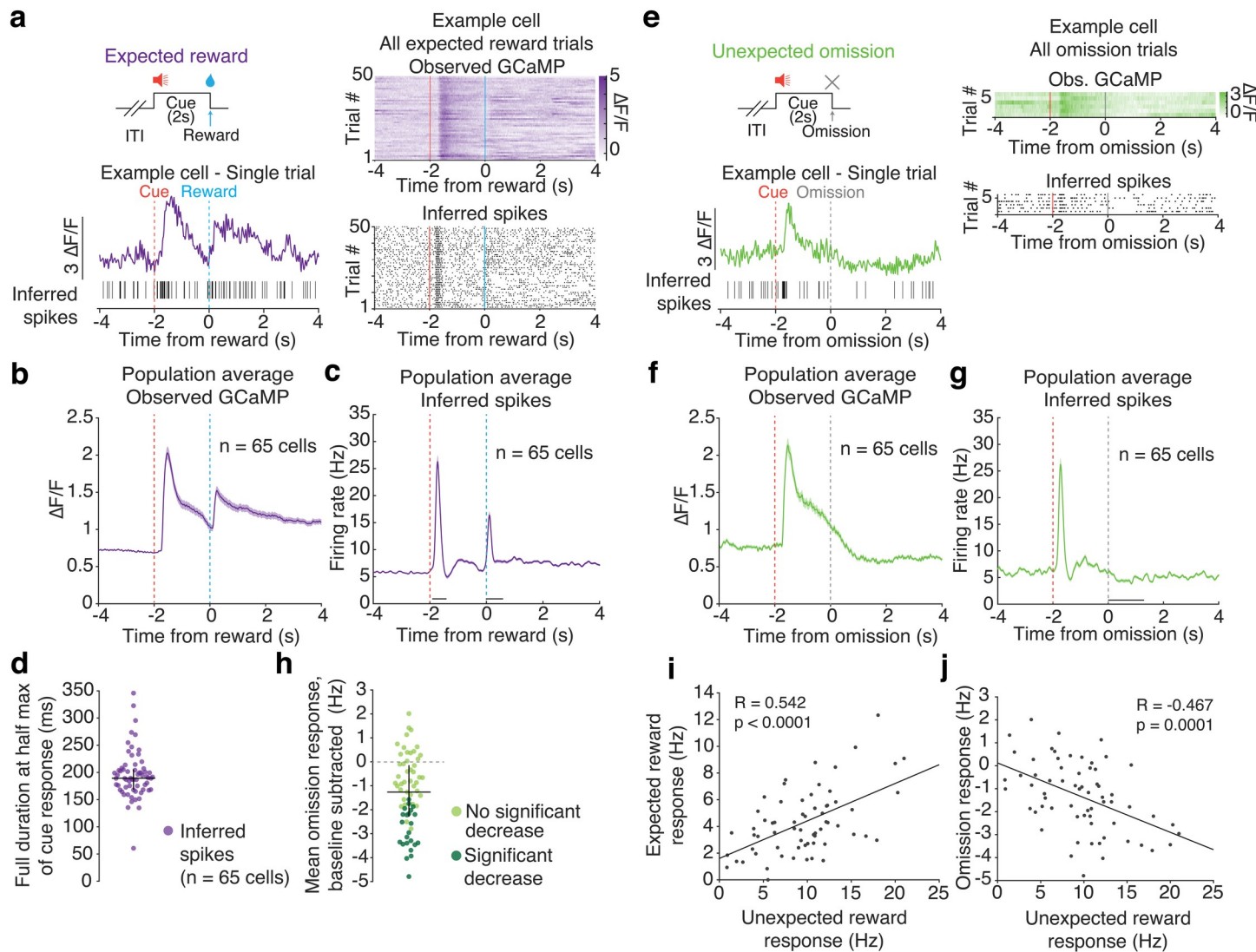

**Fig 3. Spike inference applied to Pavlovian conditioning *in vivo* imaging data recapitulates transient cue responses, pauses following reward omission, and previously reported relationships between expected and unexpected reward responses.** a. Top left: Schematic showing expected reward is delivered after a 2 s cue presentation. Bottom left: Example expected reward trial from a single cell, showing observed GCaMP (purple) and inferred spikes (black vertical lines). Right: All expected reward trials from the example cell, showing observed GCaMP (top) and inferred spikes (bottom). b. Mean observed GCaMP from population around presentation of expected reward. c. Mean inferred spikes from population around presentation of expected reward. d. Full duration at half max of cue response in inferred spikes (median = 189.3 ms, Q1 = 168.3 ms, Q3 = 205.6 ms). Vertical bars are interquartile range (Q1 and Q3). e. Top left: Schematic showing unexpected omission of reward, where reward is omitted after a 2 s cue presentation that previously predicted reward. Bottom left: Example unexpected omission trial from a single cell, showing observed GCaMP (green) and inferred spikes (black vertical lines). Right: All unexpected omission trials from example cell, showing observed GCaMP (top) and inferred spikes (bottom). f. Mean population observed GCaMP around unexpected omission of reward. g. Mean population firing rate from inferred spikes around unexpected omission of reward. Shaded areas are SEM. h. Mean omission response of inferred spikes, where omission response is mean firing rate over 1,300 ms following onset of reward omission, baseline subtracted using the mean firing rate over 1 s period before trial start (median = -1.3 Hz, Q1 = -2.3 Hz, Q3 = -0.1 Hz). Neurons that exhibited a significant decrease in firing following reward omission (22/65 neurons; 33.9% of population) are darker green. Vertical bars are interquartile range (Q1 and Q3). i. Scatterplot of expected reward response versus unexpected reward response, using inferred firing rate of each neuron, recapitulates correlations in Eshel et al. [39]. j. Scatterplot of omission response versus unexpected reward response, using inferred firing rate of each neuron, recapitulates correlations in Eshel et al. [39]. Responses in i and j are baseline subtracted. All data is from cells expressing GCaMP6f.

Thus, it is possible that the second bump of the reward response is due to receiving a larger-than-expected reward (10 μL reward, the closest volume to our experiment) [43].

The median tonic firing rate of the inferred spike dataset was similar to that of the electrophysiology dataset (inferred spike tonic firing rate median = 5.7 Hz, Q1 = 5.3 Hz, Q3 = 6.0 Hz;

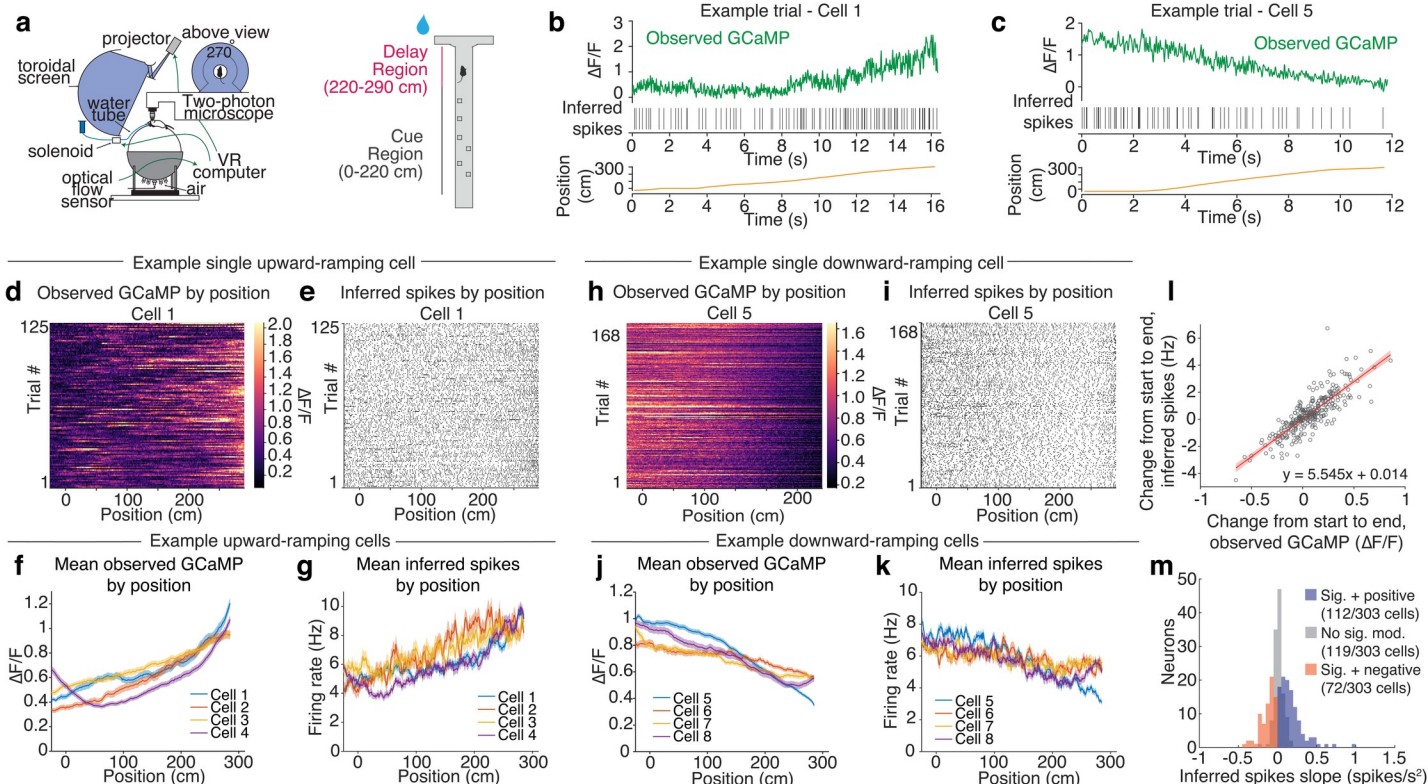

**Fig 4. Upward and downward ramps in inferred spikes during reward approach in a virtual reality environment, in agreement with recent reports from electrophysiology.** a. Left: Schematic of neural recording and behavioral setup. The mouse navigates a virtual reality (VR) environment and 2p microscope records neural data. Right: Schematic of VR T-maze paradigm. The mouse navigates a linear maze and at the end must turn to the side that featured more cues in the Cue Region to receive reward. b. Example of single trial data from an upward-ramping cell. Both observed GCaMP (green) and inferred spikes ramp upward over time as the mouse moves down the maze (position trace, yellow). c. Example of a single trial from a downward-ramping cell. Both observed GCaMP (green) and inferred spikes ramp downward over time as the mouse moves down the maze (position trace, yellow). d. All trials from the example upward-ramping cell in (b) of observed GCaMP by position. e. All trials from the example upward-ramping cell in (b) showing inferred spikes by position. f. Mean observed GCaMP by position for example upward-ramping cells. g. Mean inferred firing rate by position for example upward-ramping cells. h. All trials from the example downward-ramping cell in (c) of observed GCaMP by position. Heatmap color scales constrained to data between 1st and 99th percentile. i. All trials from the example downward-ramping cell in (c) showing inferred spikes by position. j. Mean observed GCaMP by position for example downward-ramping cells. k. Mean inferred firing rate by position for example downward-ramping cells. Shaded areas are SEM. l. Scatterplot showing how change in observed GCaMP from beginning to end of maze for each neuron relates to change in inferred firing rate for that neuron. Each data point represents a single neuron and its mean change in observed GCaMP and inferred firing rate. Red line is linear least-squares fit; shaded region is 95% confidence of the fit. m. Distribution of inferred spike slopes over time among neurons with significant position modulation with negative (red; mean slope = -0.12 spikes/s$^2$; n = 72/303 neurons) or positive (blue; mean slope = 0.18 spikes/s$^2$; n = 112/303 neurons) ramps, or with no significant position modulation (grey; mean slope = 0.02 spikes/s$^2$; n = 119/303 neurons). Significant position modulation determined by a generalized linear model (GLM), where the inferred spikes were predicted by mouse position, with a factor for individual trials, where neurons were classified as significantly ramping if the coefficient associated with position was statistically significant at level alpha = 0.01. Data is from a mix of cells expressing either GCaMP6f or GCaMP6m.

electrophysiology tonic firing rate median = 5.6 Hz, Q1 = 4.3 Hz, Q3 = 7.5 Hz; tonic firing rate calculated as the mean firing rate over a 1 s window prior to reward; Fig 2J). This implies that spike inference correctly assigns the total number of inferred spikes (which is dictated by λ) to phasic versus tonic firing. However, across neurons, the inferred spikes displayed less variability than the electrophysiology data in the tonic firing rates, presumably as a result of applying the same tuning parameter λ for all neurons, when in reality DA neurons exhibit a range of average *in vivo* firing rates between 1–10 Hz [36–38].

We found these results were robust to the specific selection of the model parameters. We repeated the spike inference for values of the tuning parameter selected to approximate 4 Hz and 8 Hz mean firing rates, both within the range of firing rates observed in dopamine neurons *in vivo* (S7A–S7E Fig), and found qualitatively similar results across this range. We also

examined the results of spike inference with decay rates γ selected as the upper and lower interquartile values from the *in vitro* experiment rather than the median value, and observed highly similar results (S8A–S8E Fig).

Taken together, our spike inference applied to *in vivo* DA imaging data closely recapitulates the median magnitude and duration of phasic activity to unexpected reward, as previously measured via electrophysiology, as well as the median level of tonic activity. However, the approach did not reproduce cross-neuron variability in tonic firing rates, and there may also be limitations to detecting a biphasic reward response.

## Application of spike inference to *in vivo* imaging data during Pavlovian conditioning

We next inferred spikes from VTA DA calcium imaging data taken during Pavlovian conditioning (Fig 3A; 2 s tone presentation followed immediately by reward delivery). This is one of the most commonly utilized behavioral paradigms to study VTA DA neuron activity [43, 44, 47–49]. Our goal was to determine whether spike inference recapitulates the previously reported features of DA firing activity during Pavlovian conditioning, including the transient, phasic response to a cue that predicts reward [50], the pause resulting from the omission of an expected reward [39, 44, 51], and finally, the previously established relationships across neurons in response to expected, unexpected, and omitted rewards [39, 51]. Because we did not have electrophysiology data from the same Pavlovian task as our imaging data, we were not able to make direct comparisons to electrophysiology for these data.

Inferred spike rates displayed a transient phasic burst to the cue onset (median full duration at half max = 189.3 ms, Q1 = 168.3 ms, Q3 = 205.6 ms), consistent with previous electrophysiology reports (Fig 3A–3D; S9 Fig) [50]. This fast, phasic response contrasted with GCaMP fluorescence, which remained elevated throughout the 2 s duration of the cue that predicted reward (Fig 3A and 3B).

We next examined whether spike inference was able to recover the decrease in tonic firing that occurs following reward omission (sometimes called "pause", Fig 3E–3H) [39, 44, 51]. Inferred spike rates in 33.9% of neurons (22/65) displayed a significant decrease in response to reward omission (Fig 3E for example neuron; Fig 3G and 3H and S10 Fig for population summaries; median omission response = -1.3 Hz, Q1 = -2.3 Hz, Q3 = -0.1 Hz; n = 65 neurons). Though a direct comparison has limitations, electrophysiology data from a Pavlovian experiment from Eshel et al. [39] following odor-reward pairings revealed that 41.9% of neurons (13/31) exhibited significant decreases in firing rate in response to reward omission (median omission response = -1.5 Hz; S11 Fig; 7 trials per cell were analyzed in order to match the available trial number in the imaging dataset). Spike inference with alternate values of the tuning parameter resulted in a slightly larger or smaller magnitude of pause response, but similar proportions of neurons with significant pause responses (S7H and S7I Fig). Spike inference with alternate decay rates, taken as estimated interquartile values from the *in vitro* estimates, generated highly similar results to inference performed using the median decay (S8H and S8I Fig). Thus, spike inference is able to recover pauses to reward omission in a similar proportion to that observed with electrophysiology, even over a range of model parameters.

Finally, it has previously been reported that across DA neurons, responses to unexpected reward are positively correlated with responses to expected reward, while unexpected reward responses are negatively correlated with responses to reward omission [39, 51]. We recovered both of these relationships on the basis of inferred spikes (Fig 3I and 3J), even when using alternate tuning parameters (S12 Fig) or decay rates (S13 Fig). These relationships were not reliably recovered directly from GCaMP fluorescence without applying spike inference (S14

Fig). This is because, as mentioned above, the fluorescence in response to the cue has not returned to baseline at the time of the reward or omission (Fig 3B and 3F). Therefore, accounting for this baseline offset through spike inference avoids inaccuracies in response estimates at the time of reward.

### Application of spike inference to *in vivo* DA calcium imaging data collected during a task in virtual reality

In addition to the phasic modulation of DA activity in response to reward or reward-predicting cues, there are an increasing number of reports suggesting that activity may "ramp" as animals approach a reward in space [22, 52–56]. Most of these studies were based on calcium imaging [22, 56], or on measures of downstream DA release [52, 53, 55], but more recently evidence has emerged from *in vivo* electrophysiology of ramps in the tonic firing of VTA DA neurons during reward approach [54].

Thus, a key question is whether spike inference recovers the recently observed upward and downward ramps in tonic DA neuron firing. To our knowledge, spike inference has not previously been validated in the context of tonic firing rate modulations in any cell type.

To this end, we applied our spike inference procedure to a recent virtual reality T-maze study with 2-photon imaging of DA neurons (Fig 4A) [22]. Again, for these spike estimates, the decay$\gamma$ was set to the median decay rate from the *in vitro* measurements at 37˚C ($\gamma$ = 0.970 for GCaMP6f at 37˚C; $\gamma$ = 0.984 for GCaMP6m at 37˚C at 60 Hz sampling frequency), and the tuning parameter $\lambda$ was set to correspond to a mean firing rate of 6 Hz for each neuron. Ramps in both GCaMP fluorescence and inferred spikes were often evident on single trials (Fig 4B–4I). Example neurons with prominent downward or upward ramps in GCaMP fluorescence showed corresponding ramps in inferred firing rates consistent with modulation of tonic activity (where tonic activity is defined as a firing rate between 1–10 Hz; Fig 4F–4K). Across the population of 303 neurons, there was a strong correlation in the extent (and direction) of change in the fluorescence signal and the inferred spike rates across the maze (correlation coefficient R = 0.832; p = 6.9x10$^{79}$; Fig 4L), with a change in ΔF/F of 0.25 from the beginning to the end of the maze associated with a change in firing rate of ~1.4 Hz (Fig 4L). Neurons with significantly positive ramps in inferred firing rates increased in firing by an average of 0.18 spikes/s$^2$ (n = 119/303 neurons; Fig 4M), whereas those with significant negative ramps decreased in firing by an average of 0.12 spikes/s$^2$ (n = 72/303 neurons; Fig 4M). The proportions of ramping neurons and ramp magnitudes were similar across a range of tuning parameters and decay rates (S15 and S16 Figs). These changes in firing rate across time are comparable in magnitude and distribution to those observed in a recent study that used electrophysiology to record from VTA DA neurons during a different reward-approach task in virtual reality [54].

Altogether, our spike inference method predicts that the ramps in the calcium fluorescence signals during our reward-approach task in VR are mediated by modest ramps in the tonic firing rates of these neurons. The ramps in our inferred firing rates are well predicted by the magnitude of the calcium ramps, and are consistent with those measured in a recent study that performed electrophysiology recordings during a similar task [54]. Thus, we show that spike inference can produce the subtle, slow-timescale changes in tonic firing rates that have been reported in DA neurons as animals approach rewards.

## Discussion

Here, we establish and validate procedures to infer spikes from calcium imaging in DA neurons, making use of a recently developed spike inference algorithm [4] that depends on only

two tuning parameters, which can be selected based on knowledge of basic biophysical properties of the cell type of interest. We first validate this approach for spike inference on a new ground-truth dataset that we have collected, which consists of simultaneous calcium imaging and electrophysiology recordings in DA neurons *in vitro*. We then apply this same approach to recently published *in vivo* DA calcium imaging data during Pavlovian conditioning and virtual reality-based navigation. We find that spike inference successfully recapitulates various features of spiking in DA neurons that were previously reported based on electrophysiological measurements, including the magnitude of the phasic unexpected reward response, average tonic firing rates, the relationship between expected and unexpected reward responses during Pavlovian conditioning, pauses during the omission of expected reward, and the recently reported magnitude of upward and downward ramping tonic activity during VR-based navigation.

## Limitations of spike inference in VTA DA neurons

Despite these successes, we also noted limitations to DA spike inference *in vivo* and *in vitro*. *In vitro*, there were some missed or mis-timed spikes. *In vivo*, we did not recapitulate the observed variability in tonic firing rates, nor the smaller second peak in response to unexpected reward that has been previously reported. Since we did not have a true ground-truth dataset for the *in vivo* data, it is impossible to know whether the subtle differences we noted relative to previously published work should be attributed to shortcomings of spike inference, differences in behavioral paradigms in our imaging study versus the previous electrophysiology study, or some other factor.

One source of discrepancy between spike inference and true spikes is that we are limited to inferring spikes at double the imaging sampling frequency (60 Hz for *in vivo* data, 66.67 Hz for *in vitro* data; see Methods). While average DA neuron phasic firing rates in response to reward are typically <60 Hz (Fig 2H), firing rates can momentarily exceed this value. This may be less relevant for our *in vivo* analyses, which involved averaging firing rates across trials for a given neuron.

Another source of discrepancy could be caused by limitations in the signal-to-noise of the imaging data. While single-spike events are often visually detectable in the fluorescence trace by eye in the *in vitro* data, *in vivo* recordings tend to have lower signal-to-noise ratios that may preclude reliable detection of single-spike events [57]. Since we only considered trial-averaged properties, and since phasic DA neuron responses even to very small rewards consist of two or more spikes [39, 58], these issues may be less relevant in our data.

The fact that we did not tune the two free parameters for the spike inference algorithm per neuron in the *in vivo* data could also produce discrepancies. This might be particularly relevant for λ, given that we assumed a target firing rate of 6 Hz, whereas in reality VTA DA neuron average firing rates may range between 1–10 Hz [36–39, 46]. While we found that our spike inference results were robust to varying the tuning parameter within a reasonable range of target firing rates (S13 and S14 Figs), examining robustness of any conclusions across a range of λ is recommended in future applications of this method.

In this paper, we have considered a very simple model relating the observed fluorescence to the unobserved calcium and spike times. Like all models, ours is misspecified: in particular, our assumption that the calcium increases instantaneously due to the presence of a spike is unrealistic, as is the assumption that the error terms are independent over time. Nonetheless, this approach leads to accurate estimates of spike times. While we could fit a more complex model that avoids these two sources of misspecification, this would result in a reduction in bias at the expense of a potentially substantial increase in variance, and thus we suspect that such an approach would not lead to improved spike time estimation.

## Relationship to recent validation of spike inference in cortical neurons

Until recently, systematic comparisons of neural correlates derived from electrophysiology data and imaging data were lacking. Recent work has begun to address this gap, by comparing neural tuning properties derived from calcium imaging versus electrophysiology data in cortex [59, 60].

Siegle et al. [59] reported discrepancies in visual cortex between neural correlates based on calcium imaging versus electrophysiology, with a larger fraction of responsive neurons from electrophysiology and higher stimulus selectivity in calcium imaging, even after spike inference (using the same spike inference algorithm we apply in this paper). A forward spike-to-calcium model could not account for these discrepancies, suggesting that it was not due to the efficacy of spike inference. Instead, the discrepancy seemed to be caused by a recording bias with electrophysiology towards highly active neurons, and also inaccuracies of spike sorting. This conclusion was based on the fact that by biasing the imaging data towards highly active neurons, and also by using stricter cell sorting criteria, the discrepancies could be largely eliminated. By contrast, Wei et al. [60] found that the discrepancies they observed in tuning properties in anterolateral motor cortex between imaging and electrophysiology could only be partially accounted for by spike inference, but could be accounted for by a forward spike-to-calcium model. This may suggest limitations in the application of their spike inference algorithms to their data.

In our case of VTA DA neuron data, as discussed above, we only noticed minor discrepancies between inferred spikes from *in vivo* imaging and previously reported electrophysiology data. The conclusions from the Siegle paper that discrepancies may be due primarily to recording biases and poorly isolated units in electrophysiology data may explain our success, as these issues may be less applicable for our data. In particular, our focus on a genetically identified cell type would help eliminate recording biases, while the phototagging used for the electrophysiology data collection requires well-isolated units. Finally, the challenge of correctly selecting the tuning parameters for spike inference, which may have been an issue for Wei et al., is less likely a problem with our spike inference algorithm, which has only two free parameters that are likely to be relatively consistent within a genetically-identified population.

In summary, we present and validate a simple and successful method to infer spikes from VTA DA neurons. We expect this straightforward approach to generalize to other subpopulations of genetically identified subcortical neurons.

## Methods

### Animals and surgery

All experimental procedures were conducted in accordance with the National Institutes of Health guidelines and were reviewed by the Princeton University Institutional Animal Care and Use Committee (IACUC). A total of 41 mice were used across all experiments. We used either DAT::IRES-Cre mice (n = 17, The Jackson Laboratory strain 006660; extensively characterized in [61]) or mice resulting from the cross of DAT$^{IREScre}$ mice and the GCaMP6f reporter line Ai148 mice [62] (n = 24, Ai148xDAT::cre, The Jackson Laboratory strain 030328).

For the slice recording experiments, we used male and female Ai148xDAT::cre mice (n = 10) and DAT::IRES-Cre mice (n = 3). For the Pavlovian conditioning experiments, male and female Ai148xDAT::cre mice (n = 8) were used. For the virtual reality experiments, male DAT::IRES-Cre mice (n = 14) and male Ai148xDAT::cre mice (n = 6) were used.

Mice were maintained on a 12-hour light on– 12-hour light off schedule. All procedures were conducted during their light off period. Mice were 2–6 months old.

For Pavlovian and virtual reality experiments, mice between 8–12 weeks underwent sterile stereotaxic surgery under isoflurane anesthesia (3–4% for induction, 0.75–1.5% for maintenance). The skull was exposed and the periosteum removed using a delicate bone scraper (Fine Science Tools). The edges of the skin were affixed to the skull using a small amount of Vetbond (3M). We injected 800 nL of a viral combination of AAV5-CAG-FLEX-GCaMP6m-WPRE-SV40 (n = 12) or AAV5-CAG-FLEX-GCaMP6f-WPRE-SV40 (n = 2; UPenn Vector Core) with $1.6x10^{12}$/mL titer and AAV9-CB7-CI-mCherry-WPRE-rBG (UPenn Vector Core) with $2.3x10^{12}$/mL titer. Two such injections were made at stereotactic coordinates: 0.5 mm lateral, 2.6 or 3.8 mm posterior, 4.7 mm in depth (relative to bregma). After the injections, we implanted a 0.6 mm diameter GRIN lens (GLP-0673, Inscopix or NEM-060-25-10-920-S-1.5p, GrinTech) in the VTA using a 3D printed custom lens holder. After implantation, a small amount of diluted metabond cement (Parkell) was applied to affix the lens to the skull using a 1 ml syringe and 18 gauge needle. After 20 minutes, the lens holder grip on the lens was loosened while the lens was observed through the microscope used for surgery to ascertain there was no movement of the lens. Then, a previously described titanium headplate was positioned over the skull using a custom tool and aligned parallel to the stereotax using an angle meter [2]. The headplate was then affixed to the skull using metabond. A titanium ring was then glued to the headplate using dental cement blackened with carbon. Imaging data was collected beginning 3–4 weeks after virus injection surgeries.

For *ex vivo* recordings of GCaMP6m, mice between 6–7 weeks underwent stereotaxic surgery under isoflurane anesthesia (3–4% for induction, .75–1.5% for maintenance). The skull was exposed and the periosteum removed using a delicate bone scraper (Fine Science Tools). The edges of the skin were affixed to the skull using a small amount of Vetbond (3M). We injected 500 nL of AAV5-CAG-FLEX-GCaMP6m-WPRE-SV40 (n = 3) with $1.6x10^{12}$/mL titer. Injections were made at stereotactic coordinates: 0.5 mm lateral, 3.2 mm posterior, 4.75 mm in depth (relative to bregma).

## Virtual-reality behavioral system

To enable a navigation-based decision-making task under head-fixed conditions, we used a virtual-reality system similar to that previously described [22, 63, 64] (Fig 4A). Mice were held head-fixed under a two-photon microscope using two custom headplate holders and ran on an air-supported, Styrofoam spherical treadmill that was 8 inches (20.3 cm) in diameter. The sphere's movement were measured using an optical flow sensor (ADNS3080) located underneath the sphere and controlled by an Arduino Due; this information was sent to the virtual-reality computer, running the ViRMEn software engine [65] (https://pni.princeton.edu/pni-software-tools/virmen) under MATLAB, which displayed and controlled the virtual-reality environment.

The display was projected using a DLP projector (Mitsubishi HD4000) running at 85 Hz onto a custom toroidal screen with a 270˚ horizontal field of view. Reward delivery was accomplished by sending by a TTL pulse from the virtual-reality computer to a solenoid valve (NResearch), which released a drop of a water to a lick tube located slightly in front and below the mice's mouth. The tone signifying trial failure was played through conventional computer speakers (Logitech). The setup was enclosed in a custom-designed cabinet built from optical rails (Thorlabs) and lined with sound-absorbing foam sheeting (McMaster-Carr).

## *In vivo* optical imaging and data acquisition

Imaging was performed using a custom- built, virtual-reality-compatible two-photon microscope [64]. The microscope was equipped with a pulsed Ti:sapphire laser (Chameleon Vision,

Coherent) tuned to 920 nm. The scanning unit used a 5-mm galvanometer and an 8-kHz resonant scanning mirror (Cambridge Technologies). The collected photons were split into two channels by a dichroic mirror (FF562-Di03, Semrock). The light for the respective green and red channels was filtered using bandpass filters (FF01-520/60 and FF01-607/70, Semrock), and then detected using GaAsP photomultiplier tubes (1077PA-40, Hamamatsu). The signal from the photomultiplier tubes was amplified using a high-speed current amplifier (59–179, Edmund). Black rubber tubing was attached to the objective (Zeiss 20×, 0.5 NA) as a light shield covering the space from the objective to the titanium ring surrounding the GRIN lens. Double-distilled water was used as the immersion medium. The microscope could be rotated along the mediolateral axis of the mice, allowing alignment of the optical axes of the microscope objective and GRIN lens as previously described for microprism imaging [64]. Control of the microscope and image acquisition was performed using the ScanImage software (Vidrio Technologies) that was run on a separate (scanning) computer. Images were acquired at 30 Hz at a resolution of 512 × 512 pixels. Average beam power measured at the front of the objective was 40–60 mW. Synchronization between the behavioral logs and acquired images was achieved by sending behavioral information each time the virtual-reality environment was refreshed from the virtual-reality computer to the scanning computer via an I2C serial bus; behavioral information was then stored in the header of the image files.

## Behavioral training

Seven days after the surgery, mice were started on a water restriction protocol that was reviewed and approved by the Princeton University IACUC. All mice received a daily allotment of water of 1–1.5 mL. Mice received water as rewards during the behavioral session. If a mouse did not receive at least 1 mL in the session, at the end of the day the mouse was given water to complete the 1 mL/day allotment. Mice received at least 1 mL of water per day but could receive up to a maximum of 1.5 mL of water per day based on performance during the T-maze task. Mice were monitored for signs of dehydration, distress, or reductions in body mass below 80% of the initial value. If any of these conditions occurred, mice were given ad libitum access to water until recovery. All mice had reduced body mass but remained above 85% of their baseline weight. Two mice showed signs of dehydration and were removed from the study. Both of these mice recovered. No mice showed signs of distress as a result of water restriction. No mice died due to water restriction. The animals were handled daily from the start of water restriction. Five days after starting water restriction and handling, mice began training in the behavioral setup. Training consisted of a shaping procedure with nine levels of T-mazes with progressively longer stem length and cognitive difficulty. After shaping concluded, in each session the first few trials (5–30) were warm-up trials drawn from mazes 5–8, and then trials from the final maze (#9) were used for the remainder of the session. Warm-up trials were excluded from all analyses presented in the paper. The mice typically received their daily allotment of water during task performance; if not, the remainder was provided to them at the end of the day.

## Details of the virtual reality behavioral task

At the beginning of each trial, mice were presented with the start of a virtual T-maze. After 30 cm (start region; -30 cm to 0 cm in Fig 4) the cue region began, in which cues randomly appeared on either side of the corridor. The portion of the maze where cues were presented (cue region) was 220 cm long (0 cm to 220 cm in Fig 4), and after it the stem of the T-maze continued for another 80 cm in which no cues were presented (delay region; 220 cm to 300 cm in Fig 4). At the end of the T-maze, the mouse had to enter one of the arms. Turning into the

correct (more cues) side would elicit a water reward (6.4 μL), whereas an incorrect choice elicited a tone (pulsing 6–12 kHz tone for 1 s). At the time of reward or tone delivery, the visual environment froze for 1 s, and then disappeared for 2 s (after a successful trial) or 5 s (after a failed trial) before another trial was started. Cue distribution is described in detail in our previous methods [22].

## Pavlovian conditioning and unexpected reward behavioral paradigm

After water restriction and handling, mice were habituated to head fixation for 2–3 sessions. Training consisted of 5 sessions (1 session per day); each session consisted of 50 reward deliveries (8 μL of water reward). During training, each reward was preceded by a 2 s tone that ended at the time of reward delivery. The time between a reward and the next tone delivery was sampled from an exponential distribution with a mean of 40 s. The tone consisted of a sum of multiple sine waves with frequencies of 2, 4, 6, 8 and 16 kHz, and an amplitude of 70 dB. All of the mice exhibited anticipatory licking by the end of the five days (increase in lick rate after tone presentation but before reward delivery). Some of the mice were previously trained for several days in a similar protocol in which the tone amplitude was 60 dB and the time between reward and subsequent tone was sampled from a uniform distribution between 5 and 15 s; these mice did not exhibit anticipatory licking until trained in the final protocol. After training, mice underwent a single test session that consisted of 64 trials; 50 of those trials were identical to the training trials (tone followed by reward), 7 trials were unexpected reward trials (reward delivery with no preceding tone) and 7 trials were unexpected omissions (tone not followed by reward). In all cases, the intertrial interval was sampled from an exponential distribution with a mean of 40 s. Trial identity was sampled randomly with the following exceptions: (1) the first five trials were standard trials (tone plus reward); (2) the first two non-standard trials were unexpected reward trials.

## Unexpected reward paradigm from Eshel et al.

We used previously collected in vivo electrophysiology data as a comparison to our inferred spike data during unexpected delivery of reward [39]. After >1 week of recovery from stereotactic surgery involving implantation of a headplate and microdrive containing optrodes, mice were water-restricted. Weight was maintained above 90% of baseline body weight. Animals were head-restrained and habituated for 1–2 d before training. Mice underwent a variable-reward task, where mice received either water reward that was either predicted by odor delivery ("trial type 1"; 45% of all trials), or where water rewards of various sizes were delivered without any odor ("trial type 2"; 45% of all trials). For both experiments, reward sizes were chosen pseudorandomly from the following set: 0.1, 0.3, 1.2, 2.5, 5, 10, or 20 μL. For our comparison, we only looked at unexpected delivery of 10 μL rewards. We used these trials because they most closely matched the conditions of our unexpected reward trials (where unexpected 8 μL water rewards were delivered in a similar head-fixed protocol). We used electrophysiology recorded extracellularly from VTA during this task. Electrophysiology methods were previously described in detail [39].

## Motion correction procedure for *in vivo* fluorescence data

Deep brain imaging can be associated with spatially non-uniform fast motion (frame to frame), as well as spatially non-uniform slow drift of the field of view (over several minutes). To perform accurate motion correction despite the spatial non-uniformity, we divided the video into small regions ("patches") that had relatively uniform motion, and separately corrected the motion within each patch, as described below. Motion correction was performed on

the red channel of the recording when available, otherwise it was performed on the green channel ($n$ = 9).

As described previously [22], before dividing the video into patches, we first performed rigid motion correction using a standard normalized cross-correlation method, to eliminate any spatially uniform motion ('matchTemplate' function in the openCV package in Python). This correction was performed on non-overlapping 50 s video clips to eliminate concerns that slow drift over the course of minutes would degrade performance. The template for the cross-correlation was calculated by dividing each clip into non-overlapping sections of 100 frames, calculating the mean image of each section and obtaining the median of the mean images. Before these motion correction steps, the video was pre-processed as follows: (i) thresholded by subtracting a constant number and setting negative values to 0, such that the lower roughly 50% of pixels was 0; (ii) using the openCV function 'erode' (with a scalar '1' kernel); and (iii) convolved with a Gaussian (s.d. = 2 pixels). Motion correction and template calculation were performed iteratively ten times or until all absolute shifts were less than one pixel in both axes. Finally, the 50 s clips had to be aligned to each other. This required generating a 'master template' for the entire video, and then using the same normalized cross-correlation procedure as before ('matchTemplate' function). The master template was calculated by taking the median of the templates of all clips.

The next step of motion correction involved compensating for spatially non-uniform, slow drift by estimating the drift in local patches. Patches were defined manually around neurons of interest to contain objects that drifted coherently (patch width 80–160 pixels). To estimate the drift of each patch over time, we used a non-rigid image registration algorithm (demons algorithm, 'imregdemons' function in MATLAB). This algorithm outputs a pixel-by-pixel correction. However, direct application of this correction risks distorting the shape of the neurons or the amplitude of the signals. Therefore, we applied a uniform correction for each patch, based on the average shift of all pixels in the patch (based on the demons output). We implemented the demons algorithm on the templates from the 50 s clips described in the previous paragraph, again using the median of these templates as the master template. The registration and master templates were computed iteratively 20 times, or until the increase in the average correlation between each corrected template and the overall template was less than the s.e.m. of these correlations. We found that the performance of the non-rigid registration improved if the templates were first processed through a local normalization procedure [66].

Finally, we performed standard rigid motion correction using the normalized cross-correlation method on each patch and clip. We then repeated the rigid motion correction after taking a rolling mean of every two frames and downsampling the video by a factor of two. This increased the signal strength; we used this downsampled video for subsequent analysis. After correcting for motion within clips, we had to correct across clips. To this end, we performed rigid motion correction on the clip templates. The motion correction code can be found in: https://github.com/benengx/Deep-Brain-Motion-Corr.

### *In vitro* recordings to compare GCaMP6 fluorescence with electrophysiology in DA neurons

In order to compare GCaMP6f and GCaMP6m fluorescence with spike times in DA neurons, we performed *ex vivo* slice imaging and electrophysiological recordings in Ai148xDAT::Cre mice or DAT::IRES-Cre mice virally expressing GCaMP6m. Mice were anesthetized with an i. p. injection of Euthasol (0.06ml/30g) and decapitated. After extraction, the brain was immersed in ice-cold carbogenated NMDG ACSF (92 mM NMDG, 2.5 mM KCl, 1.25 mM NaH2PO4, 30 mM NaHCO3, 20 mM HEPES, 25 mM glucose, 2 mM thiourea, 5 mM Na-ascorbate, 3 mM Na-pyruvate, 0.5 mM CaCl2·4H2O, 10 mM MgSO4·7H2O, and 12 mM

N-Acetyl-L-cysteine) for 2 minutes. The pH was adjusted to 7.3–7.4. Afterwards coronal slices (300um) were sectioned using a vibratome (VT1200s, Leica) and then incubated in NMDG ACSF at 34˚C for 15 minutes. Slices were then transferred into a holding solution of HEPES ACSF (92 mM NaCl, 2.5 mM KCl, 1.25 mM NaH2PO4, 30 mM NaHCO3, 20 mM HEPES, 25 mM glucose, 2 mM thiourea, 5 mM Na-ascorbate, 3 mM Na-pyruvate, 2 mM CaCl2·4H2O, 2 mM MgSO4·7H2O and 12 mM N-Acetyl-l-cysteine, bubbled at room temperature with 95% O2/ 5% CO2) for at least 45 mins until recordings were performed.

During cell-attached recordings, slices were perfused with a recording ACSF solution (120 mM NaCl, 3.5 mM KCl, 1.25 mM NaH2PO4, 26 mM NaHCO3, 1.3 mM MgCl2, 2 mM CaCl2 and 11 mM D-(+)-glucose, continuously bubbled with 95% O2/5% CO2) held at either 30˚C or 37˚C. Picrotoxin (100 μM) was added to the recording solution to block tonic inhibition and promote spontaneous activity. Cell-attached recordings were performed using a Multi-clamp 700B (Molecular Devices, Sunnyvale, CA) using pipettes with a resistance of 4–6 MOhm filled with a solution identical to the recording ACSF. Infrared differential interference contrast–enhanced visual guidance was used to select neurons that were 3–4 cell layers below the surface of the slices, which were held at room temperature while the recording solution was delivered to slices via superfusion driven by a peristaltic pump. Cell-attached recordings were collected once a seal (>50 MOhm to ~1 GOhm) between the recording pipette and the cell membrane was obtained. Action potential currents were recorded in voltage-clamp mode with voltage clamped at 0 mV. Cell-attached currents were low-pass filtered at 1 kHz and digitized and stored at 10 kHz (Clampex 9; MDS Analytical Technologies). All experiments were completed within 4 hours after slicing the brain. Fluorescence was imaged using a CMOS camera (ORCA-Flash 2.8, Hamamatsu) at 33.333 Hz (30 ms exposure windows) using a GFP filter cube set (exciter ET470/40x, dichroic T495LP, emitter ET525/50m).

## Motion correction procedure for *in vitro* data

On our *in vitro* imaging data, we performed rigid motion correction using a standard normalized cross-correlation method, to eliminate any spatially uniform motion ('matchTemplate' function in the openCV package in Python). The template for the cross-correlation was hand-selected as a frame near the middle of the recording when cell fluorescence was relatively bright. Since these recordings were usually around 2 mins in length, we performed motion correction on the entire recording rather than sectioning the recording into individual "patches."

## ROI selection and calculation of raw GCaMP for in vivo and in vitro data

As previously described [22], to determine ROIs for fluorescence measurement, we used the mean (*in vivo* only) or S.D. (*in vivo* and *in vitro*) projection of the recording to generate a clear outline of the cell body. ROIs were defined manually using this S.D. projection. An initial automatic annulus was generated by enlarging the borders of the ROI twice (by 5 μm and 10 μm); the annulus was the shape contained between the two enlarged borders, where we expect that observed activity would be due to the neuropil but not the cell itself. Next, we manually reshaped the annulus region to avoid any visible dendrites, processes or cell bodies, while approximately maintaining its original area.

To correct for neuropil contamination in the *in vivo* and *in vitro* imaging data, we subtracted a scaled version of the annulus fluorescence from the raw trace: $F_{corr}(t) = F_{raw}(t) - Y^*F_{annulus}(t)$, in which $F_{raw}(t)$ is the mean fluorescence in the ROI of each neuron at time $t$; $F_{annulus}(t)$ is the mean fluorescence in the corresponding annulus ROI at time $t$; and $Y$ is the correction factor [1, 67]. The correction factor is intended to reflect the fraction of the $z$-section that is generated by the neuropil versus the cell that is being imaged. The correction factor used was 0.58 for the

*in vivo* data and 1 for the *in vitro* data. For the *in vitro* data, spike time estimates and decay rate estimates were similar whether a correction factor of 1 or 0.7 was used.

### *In vitro* data organization

The *in vitro* data consist of 12 recordings of GCaMP6f at 30˚C; 24 recordings of GCaMP6f at 37˚C; and 21 recordings of GCaMP6m at 37˚C. Each recording consists of a time series of fluorescence intensities obtained via calcium imaging at 33.33 Hz as well as a set of ground-truth spike times obtained via an electrode. We defined bursting as a series of at least 3 spikes with a maximum inter-spike interval of 100 ms. Because we anticipated burst activity in the *in vivo* experiments, we focused our analysis on recordings where the fraction of spikes within a burst was > = 10%. Recordings where the fraction of spikes within a burst was <10% were deemed pacemaker cells and were excluded from analysis.

### *In vitro* data preprocessing

We performed data preprocessing to calculate ΔF/F and to further adjust for changes in baseline fluorescence.

First, we shifted and scaled the fluorescence $F_t$ by the baseline fluorescence $F_t^0$ to produce a normalized trace, $G_t = \Delta F_t / \hat{F}_t^0 = (F_t - \hat{F}_t^0))/\hat{F}_t^0$, where $\hat{F}_t^0$ is an estimate of the baseline fluorescence at $t$. We estimated $\hat{F}_t^0$ by the 8% percentile of $F_{t-L}, F_{t-(L-1)}, \ldots, F_t$ where $L=60s\times33.33Hz$, corresponding to a 60 second lag.

To further remove linear trends that may occur over the course of an entire fluorescence recording, we computed the residuals from a robust linear regression of $G_t$ onto time with Gaussian mixture errors, as in Theis et al. [18].Then, we subtracted the 1st percentile from these residuals, and divided by the difference between the 80th and 1st percentiles. We let $\check{G}_t$ denote the resulting scaled and shifted residuals.

Next, to allow for the possibility that more than one spike occurs within a (1/33.33)s window, we resampled the data $\check{G}_t$ from 33.33 Hz to 66.67 Hz using the scipy.signal.resample function in python. We call this time series the fluorescence trace, or the GCaMP ΔF/F, and denote it by $y_t$ for $t = 1, \ldots, T$ time steps, where now the length of each time step is 1/66.67 seconds.

### *In vitro* data analysis

**Statistical model.**    In order to infer spike times on the basis of the calcium imaging data, we consider a model for calcium dynamics proposed by [14] and [8]. We view the fluorescence trace $y_t$ at the $t$th time step as a noisy observation of the underlying calcium concentration $c_t$, which is assumed to decay with rate $0<\gamma<1$, unless there is a spike. A spike causes an instantaneous increase in the calcium concentration. The full model is

$$y_t = c_t + \epsilon_t, \quad \epsilon_t \overset{\text{iid}}{\sim} \mathcal{N}(0, \sigma^2), \quad t = 1, \ldots, T,$$

$$c_t = \gamma c_{t-1} + z_t, \quad t = 2, \ldots, T,$$

where $z_t = 0$ unless there is a spike at the $t$th timepoint, in which case $z_t>0$.

As previously discussed [4, 5], a natural estimate of the spike times, i.e. the times $s$ such that $z_s>0$, is obtained by solving the non-convex optimization problem

$$\underset{c_1, \ldots, c_T}{\text{minimize}} \left\{ \frac{1}{2} \sum_{t=1}^{T} (y_t - c_t)^2 + \lambda \sum_{t=2}^{T} 1_{(c_t \neq \gamma c_{t-1})} \right\} \text{ subject to } c_t \geq \gamma c_{t-1}.$$

Here, $\lambda$ is a tuning parameter that controls the trade-off between the goodness of fit to the data—as measured by the residual sum of squares between estimated calcium and the observed fluorescence—and the number of inferred spikes. This optimization problem is non-convex but can be solved for the global optimum using a dynamic programming strategy laid out in Jewell et al. [4]. We say that we have inferred a spike if $\hat{z}_s = \hat{c}_s - \gamma\hat{c}_{s-1} > 0$. Smaller values of $\lambda$ result in more inferred spikes and thus a higher inferred firing rate (number of inferred spikes / length of recording), whereas larger values of $\lambda$ result in fewer inferred spikes, and thus a smaller inferred firing rate.

This model assumes that a spike leads to an instantaneous increase in the calcium; in reality, the calcium increase due to a spike is not instantaneous. Therefore, after fitting the model, we shift the inferred spike times by 4 time steps ($\approx 0.06 s$). To fit this model, we must specify the decay rate $\gamma$ and a value of the tuning parameter $\lambda$.

**Estimating the decay rate $\gamma$.** According to the statistical model for calcium decay, the rate of calcium decay $\gamma$ is constant across multiple inter-spike regions. Thus, given knowledge of the spiking events from the ground-truth data, we can estimate the decay rate from multiple inter-spike segments.

More precisely, given $n$ spike indices (i.e., the times where there is at least one spike) $t_1$, $t_2,\ldots,t_n$, we defined $\bar{m}(i)$ as the index between $t_i$ and $t_{i+1}$ with maximal fluorescence value, $\bar{m}(i) = \underset{t_i \leq j \leq t_{i+1}}{\mathrm{argmax}}\{y_j\}$. Similarly, we let $\underline{m}(i) = \underset{t_i \leq j \leq t_{i+1}}{\mathrm{argmin}}\{y_j\}$ be the index with minimal fluorescence value. These indices segment the fluorescence trace from peak-to-trough between spikes. Of these, we considered segments whose peak-to-trough distance is at least $K = 10$ time steps,

$$\mathcal{I} = \underset{\{i=1,\ldots,n-1\,|\,\bar{m}(i)-\underline{m}(i)>K\}}{\cup} \left(\bar{m}(i), \underline{m}(i)\right).$$

Then we estimated the decay rate by finding the value of $\gamma$ that minimizes the residual sum of squares of the best exponential curve to each of these segments,

$$\hat{\gamma} \in \underset{\gamma}{\mathrm{argmin}}\left\{ \sum_{(a,b)\in\mathcal{I}} \underset{c_0}{\min}\left\{ \sum_{t=a}^{b} \left(y_t - \gamma^{t-a}c_0\right)^2 \right\} \right\}.$$

This process is illustrated in Fig 1D. Here, values of the fluorescence trace whose indices are contained in $\mathcal{I}$ are colored blue. Using these points, $\hat{\gamma}$ is the value that gives the lowest total residual sum of squares obtained by fitting exponential curves to each set of points. For each segment, the best exponential fit with decay rate $\hat{\gamma}$ is plotted in red.

For plots showing group estimated decay rates for GCaMP sensors and temperatures (Fig 1D and 1E), we display estimated decay rates for individual neurons. In instances where we used multiple recordings from the same neuron, we calculated the neuron decay rate by taking the mean of the estimated decay rates from those individual recordings.

**Estimating the tuning parameter $\lambda$.** The tuning parameter $\lambda$ controls the tradeoff between goodness of fit to the data and the number of inferred spikes. A smaller value of $\lambda$ leads to a higher inferred firing rate, and a larger value leads to a lower inferred firing rate.

On the *in vitro* data, we set $\lambda$ so that the inferred firing rate equals the true firing rate. In greater detail, we defined $\lambda^*$ to be the value that minimizes the difference between the number of inferred spikes and the number of time bins (of duration $(1 / 66.67)s$) with at least one spike,

$$\lambda^* \in \underset{\lambda}{\mathrm{argmin}}\{|\#\{s \mid \hat{z}_s(\lambda) > 0\} - n|\},$$

where the inferred spikes $\{s \mid \hat{z}_s(\lambda) > 0\}$ were obtained by solving the optimization problem with the tuning parameter $\lambda$ and setting $\hat{z}_s(\lambda) = \hat{c}_s - \gamma\hat{c}_{s-1}$, and where $n$ is the number of time bins with at least one ground-truth spike. We efficiently computed $\lambda^*$ using binary search.

## Assessing model performance

To assess whether the inferred spikes agree with the ground-truth spikes, we used the van Rossum distance metric [40, 41]. This metric measures the distance between two discrete vectors of spike times $\mathbf{u} = (u_1, \ldots, u_n)$ and $\mathbf{v} = (v_1, \ldots, v_{n'})$ by first mapping each to a continuous space and then using the usual Euclidean distance between two functions. Specifically, a discrete vector $\mathbf{u}$ is mapped to a continuous function, $f(t; (\mathbf{u}) = \sum_{i=1}^{n} h(t - u_i)$, where $h$ is the kernel

$$h(t) = \begin{cases} 0, & t < 0 \\ e^{-t/\tau}, & t \geq 0 \end{cases}$$

and where $\tau > 0$ is a time-scale parameter. The van Rossum distance $d(\mathbf{u}, \mathbf{v}; \tau)$ between two spike trains $\mathbf{u}$ and $\mathbf{v}$ is given as the distance between the induced functions $f(t; \mathbf{u})$ and $f(t; \mathbf{v})$,

$$d(\mathbf{u}, \mathbf{v}; \tau)^2 = \frac{1}{\tau} \int_0^\infty (f(t; \mathbf{u}) - f(t; \mathbf{v}))^2 dt.$$

## *In vivo* data analysis

**Fluorescence data preprocessing.** The same data preprocessing pipeline described above in the *In vitro* data preprocessing section was used to de-trend using a robust linear regression, subtract the 1st percentile, scale by the difference between the 80th and 1st percentiles, and then upsample from 30 Hz to 60 Hz.

**Decay rate determination.** We used the decay rates estimated from the *in vitro* experiment (Fig 1D) for the *in vivo* analysis. In particular, we calculated a group median decay rate across neurons for each GCaMP indicator and temperature based on the *in vitro* data. In instances where we used multiple recordings from a single neuron, we first calculated the mean decay rate across recordings for that neuron and used that mean decay rate in the group median calculation. We adjusted these median decay rates to account for the difference in recording frequency between the *in vitro* and *in vivo* data (the latter was recorded at 60 Hz),

$$\text{decay rate at 60 Hz} = 1 - (66.67/60.00) * (1 - \text{decay rate at 66.67 Hz}).$$

The adjusted median decay rate for the appropriate GCaMP indicator ($\gamma = 0.970$ for GCaMP6f at 37°C; $\gamma = 0.984$ for GCaMP6m at 37°C) is used for all *in vivo* analysis.

**Tuning parameter determination.** We use binary search to identify a value of the tuning parameter $\lambda$ that results in a set of inferred spikes with a target average firing rate r. In particular, we solve

$$\lambda^\star \in \underset{\lambda}{\operatorname{argmin}}\{|\#\{s \mid \hat{z}_s(\lambda) > 0\} - r \cdot T/60|\}$$

where T denotes the length of the recording in time steps. For the *in vivo* analysis, our target firing rate was 6 Hz [36–39, 46].

**Analysis of firing rate data in unexpected reward and Pavlovian conditioning experiment.** For the unexpected reward and Pavlovian conditioning data (Figs 2C, 3E and 3H), peristimulus time histograms (PSTHs) were constructed using electrophysiology and inferred spike times. For a given cell and trial type, we found all spikes that occurred within some window (-4 s to 4 s) of the behavioral event. Electrophysiology spike times were organized into 1

ms time bins, whereas inferred spike times were rounded to the nearest 1 ms. For each neuron, and for each trial type, the mean instantaneous firing rate at each time bin was calculated by summing the number of spikes occurring within that 1 ms time bin across trials, and dividing by the product of the number of trials and the bin width (1/1000 ms). We then convolved this mean firing rate with a Gaussian kernel (width = 200 ms, SD = 40 ms; normalized such that the sum of the kernel equaled 1), producing a smooth PSTH for each neuron. The population firing rate PSTHs (Figs 2E–2F, 3C and 3G), was calculated as the average of the individual cells' smooth PSTHs.

For a given cell, mean firing rates for a given period were calculated by taking the total number of spikes in some time window, and dividing by the product of the width of that window and the number of relevant trials. We calculated the baseline firing rate over a 1 s period immediately before reward delivery (for unexpected reward trials) or cue onset (for expected reward and reward omission trials). Cue, reward, and omission responses were baseline-subtracted using the baseline firing rates calculated across those respective trials. The time windows for cue, reward, and reward omission responses were selected for consistency with the Eshel et al. experiments [39]. The responses to unexpected and expected reward were calculated by finding the mean firing rate over a 600 ms window after reward onset. The response to cue presentation was calculated by finding the mean firing rate a 500 ms window beginning 150 ms after cue onset. The response to reward omission was calculated by finding the mean firing rate over a 1,300 ms window after reward omission onset. Peak amplitudes of cue and reward responses were calculated as the largest peak of that cell's smooth PSTH in the respective response window. Full duration at half max of cue and reward responses were calculated using the width of the largest peak, as measured at half the max value, of the smooth PSTH in the response window.

To detect significant changes in response to reward omission for a given neuron, for each trial we calculated the firing rate in a 1 s window prior to cue onset and the firing rate in a 1,300 ms window following omission onset. We performed a paired t-test to determine whether the firing rates during the omission window were significantly different from baseline, and categorized a neuron as having a significant omission response if the t-test rejected the null hypothesis at level $p = 0.05$ and the mean omission response was $< 0$ Hz.

**Analyzing GCaMP fluorescence and inferred spike rates by maze position in the VR task.** In Fig 4, we plotted GCaMP fluorescence and inferred spike rates as a function of position. To smooth this data for visualization (Fig 4G and 4K), for a given cell we calculated moving averages of GCaMP fluorescence and inferred spike rate using a 10 cm window moved at 1 cm intervals along the maze. To compute the inferred spiking rate by position, we divided the number of inferred spikes within a given 10 cm window by the amount of time spent in that window. We then averaged GCaMP and inferred spike rates across trials to generate cell averages (Fig 4G and 4K).

**Quantifying ramps in VR task using GCaMP fluorescence and inferred spikes.** To describe linear ramps by position in each neuron (Fig 4L), we performed a linear regression of the observed GCaMP fluorescence or the inferred spike rates onto the mouse's position. We calculated the mean inferred spike rates and GCaMP fluorescence over maze positions using the same procedure for Fig 4G and 4K, except that we used non-overlapping 10 cm position bins (with edges at maze positions -30 and 290 cm), where the positional value was the center of that bin. For Fig 4L, we calculated the linear change in GCaMP or inferred spike rate for each neuron by multiplying the position coefficient of its respective linear model by the length of the maze (290 cm).

We determined whether a neuron had a significant position-related ramp by determining whether that neuron's estimated probability of spiking significantly increased or decreased as

the mouse progressed down the maze. To do this, we fit a logistic generalized linear model (GLM), where the inferred spikes were predicted by mouse position, with a factor for individual trials:

$$\log\left(\frac{p}{1-p}\right) = \beta_0 + \beta_1 \cdot \text{position} + \sum\nolimits_{i=1}^{j} \beta_i 1_{(\text{ith trial})}$$

where $p$ is the probability that a neuron spikes at a particular timestep, and $j$ is the number of trials.

We classified neurons as significantly ramping if their position significantly modulated likelihood to fire (i.e. if the coefficient associated with position is statistically significant, at level alpha = 0.01). To determine what proportion of neurons had significant positive or negative ramps, we took neurons with significant position modulation and grouped them based on the sign of the position coefficient in the logistic regression model described above.

## Supporting information

**S1 Fig. Identification of pacemaker cells by absence of phasic activity, and decay statistics for pacemaker cells.** a. Percent of spikes that occurred within a burst in a previously collected *in vivo* electrophysiology dataset (Eshel et al. [39]). A burst was defined by a series of at least 3 spikes with < = 100 ms between each spike. All *in vivo* electrophysiology cells had >10% of spikes within a burst. This was chosen as the cut-off for inclusion in *in vitro* analysis. b. Percent of spikes that occurred within a burst for all recordings in the GCaMP6f at 30˚C, GCaMP6f at 37˚C, and GCaMP6m at 37˚C groups. Cells with <10% of spikes occurring within a burst were excluded from group analysis, since they featured little or no phasic activity. c. Left: Estimated decay rates for pacemaker cells for GCaMP6f at 30˚C (n = 3 cells; median decay rate = 0.974, Q1 = 0.968, Q3 = 0.981), GCaMP6f at 37˚C (n = 7 cells; median decay rate = 0.969, Q1 = 0.963, Q3 = 0.982), and GCaMP6m at 37˚C (n = 1 cell; decay rate = 0.986). Right: Half decay time for pacemaker cells for GCaMP6f at 30˚C (median = 0.398 s, Q1 = 0.323 s, Q3 = 0.577 s), GCaMP6f at 37˚C (median = 0.332 s, Q1 = 0.278 s, Q3 = 0.566 s) and GCaMP6m at 37˚C (0.722 s).
(TIF)

**S2 Fig. Mean firing frequency in bursty cells during in vitro recordings.** Mean spike frequency during the simultaneous calcium imaging and electrophysiology *in vitro* experiment (GCaMP6f at 30˚C, n = 12 recordings from 9 total cells, cell median firing rate = 4.3 Hz, Q1 = 3.2 Hz, Q3 = 4.8 Hz; GCaMP6f at 37˚C, n = 9 recordings from 12 total cells, cell median firing rate = 6.4 Hz, Q1 = 6.0 Hz, Q3 = 9.3 Hz; GCaMP6m at 37˚C, n = 19 recordings from 9 total cells, cell median firing rate = 4.2 Hz, Q1 = 3.6 Hz, Q3 = 6.0 Hz). Vertical bars are interquartile range (Q1 and Q3).
(TIF)

**S3 Fig. Example traces from each neuron in the GCaMP6f 30˚C group.** Examples of correspondence between observed GCaMP (black line, top) and estimated calcium (blue line, top), and observed spikes (black dashes, bottom) and inferred spikes (blue dashes, bottom) for each recording in the GCaMP6f 30˚C group. Traces are 20 s long. Scale bars are 1 ΔF/F and 5 s.
(TIF)

**S4 Fig. Example traces from each neuron in the GCaMP6f 37˚C group.** Examples of correspondence between observed GCaMP (black line, top) and estimated calcium (green line, top), and observed spikes (black dashes, bottom) and inferred spikes (green dashes, bottom) for each recording in the GCaMP6f 37˚C group. Traces are 20 s long. Scale bars are 1 ΔF/F

and 5 s.
(TIF)

**S5 Fig. Example traces from each neuron in the GCaMP6m 37˚C group.** Examples of correspondence between observed GCaMP (black line, top) and estimated calcium (orange line, top), and observed spikes (black dashes, bottom) and inferred spikes (orange dashes, bottom) for each recording in the GCaMP6m 37˚C group. Traces are 20 s long. Scale bars are 1 ΔF/F and 5 s.
(TIF)

**S6 Fig. Example traces of inference results with large van Rossum distances resulting from misestimation of observed firing rate.** a-c. Examples of spike inference when tuning parameters are selected to underestimate (a), match (b), or overestimate (c) the observed firing rate of a neuron expressing GCaMP6f. d-f. Examples of spike inference when tuning parameters are selected to underestimate (d), match (e), or overestimate (f) the observed firing rate of a neuron expressing GCaMP6m. In both GCaMP6f and GCaMP6m examples, the van Rossum distance is lowest when the tuning parameter is selected to match the firing rate, but under- and overestimates appear to capture both tonic and phasic activity patterns.
(TIF)

**S7 Fig. Tuning rate comparison for Pavlovian experiment.** a. Mean population firing rates from inferred spikes around presentation of unexpected reward when λ is selected to target a 4, 6, or 8 Hz average estimated firing rate (n = 65 cells). b-e. Comparison of inferred spikes over a range of λ values and electrophysiology spikes from Eshel et al. [39]. b. Unexpected reward response, where reward response is the mean firing rate over the first 600 ms following reward presentation, baseline subtracted using the mean firing rate over a 1 s period before reward presentation (inferred spikes over baseline median = 6.5 Hz, Q1 = 3.9 Hz, Q3 = 8.2 Hz for λ target 4 Hz; median = 9.8 Hz, Q1 = 6.2 Hz, Q3 = 12.2 Hz, for λ target 6 Hz; median = 11.5 Hz, Q1 = 7.1 Hz, Q3 = 14.0 Hz, for λ target 8 Hz; electrophysiology spikes over baseline median = 9.5 Hz, Q1 = 6.3 Hz, Q3 = 12.0 Hz). c. Peak reward response amplitude in inferred and electrophysiology spikes, where peak is maximum value of PSTH in the first 600 ms period following reward presentation (inferred spikes median = 19.7 Hz, Q1 = 15.0 Hz, Q3 = 27.2 Hz, for λ target 4 Hz; median = 29.0 Hz, Q1 = 23.7 Hz, Q3 = 38.6 Hz, for λ target 6 Hz; median = 34.4 Hz, Q1 = 28.1 Hz, Q3 = 44.3 Hz, for λ target 8 Hz; electrophysiology median = 30.5 Hz, Q1 = 22.6 Hz, Q3 = 40.9 Hz). d. Full duration at half max of reward response peak in inferred and electrophysiology spikes (inferred spikes median full duration at half max = 173.4 ms, Q1 = 156.8 ms, Q3 = 226.8 ms for λ target 4 Hz; median = 183.9 ms, Q1 = 163.6 ms, Q3 = 235.9 ms for λ target 6 Hz; median = 196.2 ms, Q1 = 175.8 ms, Q3 = 242.0 ms for λ target 8 Hz; electrophysiology median full duration at half max = 153.5 ms, Q1 = 120.4 ms, Q3 = 272.5 ms). e. Mean tonic firing rates prior to presentations of unexpected reward, where tonic firing rates are calculated as the mean firing rate over a 1 s period prior to reward presentation (inferred spikes tonic firing rate median = 3.9 Hz, Q1 = 3.4 Hz, Q3 = 4.1 Hz, for λ target 4 Hz; median = 5.7 Hz, Q1 = 5.3 Hz, Q3 = 6.0 Hz, for λ target 6 Hz; median = 7.6 Hz, Q1 = 7.0 Hz, Q3 = 8.3 Hz, for λ target 8 Hz; electrophysiology tonic firing rate median = 5.6 Hz, Q1 = 4.3 Hz, Q3 = 7.5 Hz). Vertical bars are interquartile range (Q1 and Q3). f. Mean inferred spikes from population around presentation of expected reward. g. Full duration at half max of cue response in inferred spikes (median = 182.3 ms, Q1 = 168.4 ms, Q3 = 203.2 ms for λ target 4 Hz; median = 189.3 ms, Q1 = 168.3 ms, Q3 = 205.6 ms for λ target 6 Hz; median = 197.2 ms, Q1 = 174.9 ms, Q3 = 220.5 ms for λ target 8 Hz). Vertical bars are interquartile range (Q1 and Q3). h. Mean population firing rate from inferred spikes around unexpected omission of

reward. Shaded areas are SEM. i. Mean omission response of inferred spikes, where omission response is mean firing rate over 1,300 ms following onset of reward omission, baseline subtracted using the mean firing rate over 1 s period before trial start (median = -0.7 Hz, Q1 = -1.8 Hz, Q3 = -0.1 Hz, for λ target 4 Hz; median = -1.3 Hz, Q1 = -2.3 Hz, Q3 = -0.1 Hz, for λ target 6 Hz; median = -1.5 Hz, Q1 = -3.4 Hz, Q3 = -0.4 Hz, for λ target 8 Hz). Neurons that exhibited a significant decrease in firing following reward omission (21/65 neurons, 32.3% of population for λ target 4 Hz; 22/65 neurons, 33.6% of population for λ target 6 Hz; 21/65 neurons, 32.3% of population for λ target 8 Hz) are darker green. Vertical bars are interquartile range (Q1 and Q3). All imaging data is of cells expressing GCaMP6f.
(TIF)

**S8 Fig. Decay rate comparison for Pavlovian experiment.** a. Mean population firing rates (n = 65 cells) from inferred spikes around presentation of unexpected reward when λ is selected to target a 6 Hz average estimated firing rate and the decay rate γ is selected as the 25th percentile (low gamma), median, or 75th percentile (high gamma) estimated decay rate from the *in vitro* experiment. b-e. Comparison of inferred spikes over a range of γ values and electrophysiology spikes from Eshel et al. [39]. b. Unexpected reward response, where reward response is the mean firing rate over the first 600 ms following reward presentation, baseline subtracted using the mean firing rate over a 1 s period before reward presentation (inferred spikes over baseline median = 9.8 Hz, Q1 = 6.2 Hz, Q3 = 12.0 Hz for low gamma; median = 9.8 Hz, Q1 = 6.2 Hz, Q3 = 12.2 Hz, for median gamma; median = 10.3 Hz, Q1 = 6.0 Hz, Q3 = 12.0 Hz, for high gamma; electrophysiology spikes over baseline median = 9.5 Hz, Q1 = 6.3 Hz, Q3 = 12.0 Hz). c. Peak reward response amplitude in inferred and electrophysiology spikes, where peak is maximum value of PSTH in the first 600 ms period following reward presentation (inferred spikes median = 27.5 Hz, Q1 = 22.5 Hz, Q3 = 36.9 Hz for low gamma; median = 29.0 Hz, Q1 = 23.7 Hz, Q3 = 38.6 Hz for median gamma; median = 31.3 Hz, Q1 = 24.8 Hz, Q3 = 41.9 Hz for high gamma; electrophysiology median = 30.5 Hz, Q1 = 22.6 Hz, Q3 = 40.9 Hz). d. Full duration at half max of reward response peak in inferred and electrophysiology spikes (inferred spikes median full duration at half max = 190.6 ms, Q1 = 164.8 ms, Q3 = 220.3 ms for low gamma; median = 183.9 ms, Q1 = 163.6 ms, Q3 = 235.9 ms for median gamma; median = 187.6 ms, Q1 = 160.5 ms, Q3 = 223.3 ms for high gamma; electrophysiology median full duration at half max = 153.5 ms, Q1 = 120.4 ms, Q3 = 272.5 ms). e. Mean tonic firing rates prior to presentations of unexpected reward, where tonic firing rates are calculated as the mean firing rate over a 1 s period prior to reward presentation (inferred spikes tonic firing rate median = 5.6 Hz, Q1 = 5.1 Hz, Q3 = 6.0 Hz, for low gamma; median = 5.7 Hz, Q1 = 5.3 Hz, Q3 = 6.0 Hz for median gamma; median = 5.9 Hz, Q1 = 5.3 Hz, Q3 = 6.1 Hz for high gamma; electrophysiology tonic firing rate median = 5.6 Hz, Q1 = 4.3 Hz, Q3 = 7.5 Hz). Vertical bars are interquartile range (Q1 and Q3). f. Mean inferred spikes from population around presentation of expected reward. g. Full duration at half max of cue response in inferred spikes (median = 195.1 ms, Q1 = 171.9 ms, Q3 = 209.5 ms for low gamma; median = 189.3 ms, Q1 = 168.3 ms, Q3 = 205.6 ms for median gamma; median = 185.8 ms, Q1 = 165.0 ms, Q3 = 202.3 ms for high gamma). Vertical bars are interquartile range (Q1 and Q3). h. Mean population firing rate from inferred spikes around unexpected omission of reward. Shaded areas are SEM. i. Mean omission response of inferred spikes, where omission response is mean firing rate over 1,300 ms following onset of reward omission, baseline subtracted using the mean firing rate over 1 s period before trial start (median = -1.1 Hz, Q1 = -2.1 Hz, Q3 = -0.3 Hz, for low gamma; median = -1.3 Hz, Q1 = -2.3 Hz, Q3 = -0.1 Hz, for median gamma; median = -1.7 Hz, Q1 = -3.4 Hz, Q3 = -0.6 Hz, for high gamma). Neurons that exhibited a significant decrease in firing following reward omission (24/65 neurons, 36.9% of population for low gamma; 22/65

neurons, 33.6% of population for median gamma; 25/65 neurons, 38.5% of population for high gamma) are darker green. Vertical bars are interquartile range (Q1 and Q3). All imaging data is from cells expressing GCaMP6f.
(TIF)

**S9 Fig. Mean cue and expected reward responses in inferred spikes.** a. Mean cue response, where cue response is the mean firing rate in a 500 ms period beginning 150 ms after cue onset, baseline subtracted using the mean firing rate over a 1 s period before cue (median = 7.2 Hz, Q1 = 4.9 Hz, Q3 = 10.0 Hz). b. Mean expected reward response, where reward response is the mean firing rate over the first 600 ms following reward presentation, baseline subtracted using the mean firing rate over a 1 s period before cue (median = 3.8 Hz, Q1 = 2.7 Hz, Q3 = 5.6 Hz). c. Peak cue response amplitude in inferred spikes, where peak is maximum value of PSTH in the 500 ms cue period (median = 27.0 Hz, Q1 = 19.3 Hz, Q3 = 33.5 Hz). d. Peak expected reward response amplitude in inferred spikes, where peak is maximum value of PSTH in the first 600 ms period following reward presentation (median = 17.1 Hz, Q1 = 12.8 Hz, Q3 = 19.3 Hz). e. Full duration at half max of cue response in inferred spikes (median = 189.3 ms, Q1 = 168.3 ms, Q3 = 205.6 ms). f. Full duration at half max of expected reward response in inferred spikes (median = 162.0 ms, Q1 = 150.0 ms, Q3 = 199.2 ms). Vertical bars are interquartile range (Q1 and Q3). All data from cells expressing GCaMP6f.
(TIF)

**S10 Fig. Mean cue and omission responses in inferred spikes during omission trials.** a. Mean cue response during omission trials, where cue response is the mean firing rate over first 500 ms following cue presentation, baseline subtracted using the mean firing rate over a 1 s period before cue (median = 6.4 Hz, Q1 = 3.5 Hz, Q3 = 9.5 Hz). b. Peak cue response amplitude in inferred spikes during omission trials, where peak is maximum value of PSTH in the first 500 ms period following cue presentation (median = 26.3 Hz, Q1 = 19.5 Hz, Q3 = 33.0 Hz). c. Full duration at half max of cue response in inferred spikes during omission trials (median = 192.5 ms, Q1 = 176.7 ms, Q3 = 240.5 ms). d. Mean firing rate in inferred spikes over 1,300 ms period following onset of reward omission (median = 5.1 Hz, Q1 = 4.0 Hz, Q3 = 5.6 Hz). Vertical bars are interquartile range (Q1 and Q3).
(TIF)

**S11 Fig. Omission response as measured by electrophysiology during an odor cue Pavlovian task.** a. Mean population firing rate from electrophysiology spikes around unexpected omission of reward following presentation reward-predictive odor cue. Shaded areas are SEM. h. Mean omission response of electrophysiology spikes, where omission response is mean firing rate over 1,300 ms following onset of reward omission, baseline subtracted using the mean firing rate over 1 s period before trial start (median = -1.45 Hz, Q1 = -1.98 Hz, Q3 = -0.87 Hz). Neurons that exhibited a significant decrease in firing following reward omission (13/31 neurons; 41.9% of population) are darker grey. Vertical bars are interquartile range (Q1 and Q3). Only the first 7 trials for each neuron were used for determination of significant omission response to correspond to the inferred spike data. Note that this data is not used to directly compare to inferred spike modalities because of differences in behavioral paradigms, including differences in reward-predictive cue modality. Data from Eshel et al. [39].
(TIF)

**S12 Fig. Reward and omission response correlations over range of tuning parameters.** a. Scatterplot of expected reward response versus unexpected reward response recapitulates correlations in Eshel et al. [39] when using tuning parameters for a range of inferred firing rate. b. Scatterplots of omission response versus unexpected reward response recapitulates

correlations in Eshel et al. [39] when using tuning parameters for a range of inferred firing rates. Responses in a and b are baseline subtracted.
(TIF)

**S13 Fig. Reward and omission response correlations over range of decay rates.** a. Scatterplot of expected reward response versus unexpected reward response recapitulates correlations in Eshel et al. [39] when $\lambda$ is selected to target a 6 Hz average estimated firing rate and the decay rate $\gamma$ is selected as the 25th percentile (low gamma), median, or 75th percentile (high gamma) estimated decay rate from the *in vitro* experiment (Fig 1D). b. Scatterplots of omission response versus unexpected reward response recapitulates correlations in Eshel et al. [39] when using tuning parameters for a range of inferred firing rates. Responses in a and b are baseline subtracted.
(TIF)

**S14 Fig. Reward and omission response correlations using GCaMP.** a. Scatterplot of expected reward response versus unexpected reward response. Responses are calculated by finding mean $\Delta F/F$ value over a 600 ms window following reward presentation and baseline subtracting the mean $\Delta F/F$ value over a 1 s window before trial start (1 s before cue for expected reward; 1 s before reward for unexpected reward). b. Scatterplot of omission response versus unexpected reward response. Omission response is calculated by finding the mean $\Delta F/F$ value over a 1,300 ms window following reward omission and baseline subtracting the mean $\Delta F/F$ value over a 1 s window before cue.
(TIF)

**S15 Fig. Upward and downward ramps in inferred spikes during reward approach in a virtual reality environment are detectable over a range of tuning parameters.** Results of spike inference during the virtual reality T-maze task in Fig 4 when tuning parameters approximating mean firing rates of 4, 6, or 8 Hz are used. a. Mean inferred firing rate by position for example upward-ramping cells over a range of tuning parameters. b. Mean inferred firing rate by position for example downward-ramping cells over a range of tuning parameters. c. For each tuning parameter, scatterplots showing how change in observed GCaMP from beginning to end of maze for each neuron relates to change in inferred firing rate for that neuron. Each data point represents a single neuron and its mean change in observed GCaMP and inferred firing rate. Red line is linear least-squares fit; dotted region is 95% confidence of the fit. d. For each tuning parameter, histograms of changes of inferred firing rate for each neuron, color coded by whether a neuron's probability of firing significantly decreased (red), increased (blue), or did not change (grey) with increases in maze position. e. For each tuning parameter, histograms of changes of the inferred firing rate slope for each neuron, color coded by whether a neuron's probability of firing significantly decreased (red; mean slope = -0.09 spikes/s$^2$, n = 67/303 neurons for 4 Hz target; mean slope = -0.12 spikes/s$^2$, n = 72/303 neurons for 6 Hz target; mean slope = -0.15 spikes/s$^2$, n = 77/303 neurons for 8 Hz target), increased (blue; mean slope = 0.13 spikes/s$^2$, n = 102/303 neurons for 4 Hz target; mean slope = 0.18 spikes/s$^2$; n = 112/303 neurons for 6 Hz target; mean slope = 0.22 spikes/s$^2$, n = 119/303 neurons for 8 Hz target), or did not change (grey; mean slope = 0.01 spikes/s$^2$, n = 134/303 neurons for 4 Hz target; mean slope = 0.02 spikes/s$^2$; n = 119/303 neurons for 6 Hz target; mean slope = 0.02 spikes/s$^2$, n = 107/303 neurons for 8 Hz target) with increases in maze position. Significant position modulation determined by a generalized linear model (GLM), where the inferred spikes were predicted by mouse position, with a factor for individual trials, where neurons were classified as significantly ramping if the coefficient associated with position was statistically significant at level alpha = 0.01. Data includes cells expressing GCaMP6m and GCaMP6f.
(TIF)

**S16 Fig. Upward and downward ramps in inferred spikes during reward approach in a virtual reality environment are detectable over a range of decay rates.** Results of spike inference during the virtual reality T-maze task in Fig 4 when λ is selected to target a 6 Hz average estimated firing rate and the decay rate γ is selected as the 25th percentile (low gamma), median, or 75th percentile (high gamma) estimated decay rate from the *in vitro* experiment (Fig 1D). a. Mean inferred firing rate by position for example upward-ramping cells over a range of decay rates. b. Mean inferred firing rate by position for example downward-ramping cells over a range of decay rates. c. For each decay rate, scatterplots showing how change in observed GCaMP from beginning to end of maze for each neuron relates to change in inferred firing rate for that neuron. Each data point represents a single neuron and its mean change in observed GCaMP and inferred firing rate. Red line is linear least-squares fit; dotted region is 95% confidence of the fit. d. For each decay rate, histograms of changes of inferred firing rate for each neuron, color coded by whether a neuron's probability of firing significantly decreased (red), increased (blue), or did not change (grey) with increases in maze position. e. For each decay rate, histograms of changes of the inferred firing rate slope for each neuron, color coded by whether a neuron's probability of firing significantly decreased (red; mean slope = -0.12 spikes/s$^2$, n = 70/303 neurons for low gamma; mean slope = -0.12 spikes/s$^2$, n = 72/303 neurons for median gamma; mean slope = -0.14 spikes/s$^2$, n = 74/303 neurons for high gamma), increased (blue; mean slope = 0.18 spikes/s$^2$, n = 111/303 neurons for low gamma; mean slope = 0.18 spikes/s$^2$; n = 112/303 neurons for median gamma; mean slope = 0.18 spikes/s$^2$, n = 117/303 neurons for high gamma), or did not change (grey; mean slope = 0.02 spikes/s$^2$, n = 122/303 neurons for low gamma; mean slope = 0.02 spikes/s$^2$; n = 119/303 neurons for median gamma; mean slope = 0.01 spikes/s$^2$, n = 112/303 neurons for high gamma) with increases in maze position. Significant position modulation determined by a generalized linear model (GLM), where the inferred spikes were predicted by mouse position, with a factor for individual trials, where neurons were classified as significantly ramping if the coefficient associated with position was statistically significant at level alpha = 0.01. Data includes cells expressing GCaMP6m and GCaMP6f.
(TIF)

## Acknowledgments

We thank the N. Uchida laboratory for sharing electrophysiology data. We thank C. Zimmerman and J. Lee for comments on the manuscript, and members of the Witten laboratories for their support. We thank E. Engel for reagents.

## Author Contributions

**Conceptualization:** Daniela M. Witten, Ilana B. Witten.

**Data curation:** Weston Fleming, Sean Jewell, Ben Engelhard.

**Formal analysis:** Weston Fleming, Sean Jewell, Ben Engelhard.

**Funding acquisition:** Daniela M. Witten, Ilana B. Witten.

**Investigation:** Weston Fleming, Ben Engelhard.

**Methodology:** Weston Fleming, Sean Jewell, Daniela M. Witten.

**Project administration:** Daniela M. Witten, Ilana B. Witten.

**Resources:** Daniela M. Witten, Ilana B. Witten.

**Software:** Sean Jewell.

**Supervision:** Daniela M. Witten, Ilana B. Witten.

**Validation:** Weston Fleming, Sean Jewell.

**Visualization:** Weston Fleming, Sean Jewell.

**Writing – original draft:** Weston Fleming, Sean Jewell, Daniela M. Witten, Ilana B. Witten.

**Writing – review & editing:** Weston Fleming, Sean Jewell, Ben Engelhard, Daniela M. Witten, Ilana B. Witten.

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
