## [Decision Letter · Decision Letter 0]

18 Feb 2021

PONE-D-20-38050

Inferring spikes from calcium imaging in dopamine neurons

PLOS ONE

Dear Dr. Fleming,

Thank you for submitting your manuscript to PLOS ONE. After careful consideration, we feel that it has merit but does not fully meet PLOS ONE’s publication criteria as it currently stands. Therefore, we invite you to submit a revised version of the manuscript that addresses the points raised during the review process.

1. The authors do a good job of explaining their parameter estimation procedures from in vitro data, but we note that there is considerable variability in the decay rate, especially for the 6f 37C condition, where they have only 5 cells (9 recordings). That variability presents several issues. First, if the larger variability seen in that case (Fig. 1d) is real, it could be problematic for accurately estimating spiking for VTA DA neurons for this particularly common experimental condition, especially using just a single set of validated parameters (e.g. tau=1 sec, lambda=6 Hz). Second, the variability masks the ability to state whether the systematic trends in Fig. 1d between the different calcium sensors at different temperatures are significant. One would perhaps expect that the body temperature decay per time step is faster (smaller value) than for 30C, and that the 6f sensor would be faster than the 6m sensor at the same temperature. Yet the authors are not able to conclude that given the variability. We thus recommend doubling the number of cells, so it is comparable to that of the other two conditions.

2. Related to the above, we wonder how much error is introduced by the variation in the decay per time step when only the median or mean value of the (small) population is used for all cells. For example, might there actually be two different subpopulations of DA neurons (e.g. tonic and bursty), so that the median/mean does not do well for one or the other? Consistent with that, Fig. 1f (middle) seems to show some particularly large spike estimation errors for the 6f/37C case, even when the firing rate is matched to the electrophysiologically observed rate. The cost of misestimating the decay rate would be helpful to understand, and the authors should generate a similar set of curves as Fig. 1f except with the decay rate along the x-axis.

3. For several neurons/recordings, the estimation tends to predict higher calcium levels than observed in the troughs between bursts – never lower. Can the authors comment on why there is such a systematic error, and perhaps a discussion on what could be done to improve the model (or further caveats on the current model’s limitations)?

4. Although Data is fully available without restrictions, there is no indication of where to obtain the data. According to PLoS requirements, availability upon request is insufficient and some information on where the data can be found (e.g. repository) is recommended.

5. Although the authors acknowledge that the decay and rate parameters vary across neurons (and therefore the main utility of the model is in interpreting population level dynamics rather than single neurons), it is unclear from the data presented how much the parameters actually vary across neurons and how small changes in the parameter estimates might impact inferred spiking across the population. This point seems critical for determining whether the parameter estimates given can be broadly applied to dopamine neurons in-vivo, especially given the significant variability observed in other regions and cell-types. Specifically:

5.1.     Decay parameter medians and ranges for in-vitro data are shown in Figure 1, but the authors should include data points for each individual neuron, especially given that the number of neurons for each condition is quite low (n’s should be listed in the legend).

5.2.     Each ‘recording’ in Figure 1 was considered an independent measurement even though two recordings were made for some neurons (lines 59-61). It’s not clear whether separate recordings from the same neurons were used to calculate the median decay times used for the in-vivo inference.

5.3.     It would be useful to determine how varying the rate and decay across neurons within some range (ideally estimated from in-vitro recordings), impacts estimates of phasic increases and decreases and slow ramps. Perhaps small changes would not change the population spiking estimates much, but this seems important to know when applying to future in-vivo recordings which may sample from different dopamine neuron populations.

In Figure 4, the authors show that positive and negative ramping spike rates can be recovered from ramping calcium signals in single neurons using their model. However, nothing is shown to validate their inference against electrophysiological data as in Figures 2 and 3. Does their spike inference model reveal features of the ramping that cannot be gleaned from the calcium signal alone (as for reward omission trials)? Other studies (i.e. Wei et al 2020) have argued that negative and positive ramps in single neuron spiking are differently represented in calcium signaling – does this model reconcile those discrepancies? Since the model parameters can vary across single neurons, can the authors show features of spike ramping at the population level that can be accurately recovered from their model (e.g. rates or proportions of positive and negative ramping neurons?)

Other points

a) In Fig. 1f, a tau value of 1 sec is chosen for all the curves. How much improvement is there is the actual fitted tau for each cell is used?

b) Also as a supplement for Fig. 1f,  it would be helpful to the reader to illustrate what a difference of 5000 in the van Rossum distance means for the raster plot

c) In the Methods, it is not clearly stated how long after the virus injection surgery the authors waited before performing imaging studies.

d) In the Methods, the authors comment on the need to precisely align the animal on the sphere. The “custom alignment tool” referred to is not particularly transparent. For reproducibility by others, clarifying what this tool is and how it is used would be useful.

e) To account for the rise time kinetics of the indicator, the authors state that they shift the inferred spike times by 4 time steps: some discussion of whether this shift is consistent among the indicators or neurons might be useful, along with a discussion on how this shift might be tuned, and why it is optimal to consider this potential third free parameter in this way rather than build in a double exponential with a rise parameter, as is often used to model these calcium transients

f) The type of GCaMP (m or f) used for imaging experiments should be indicated in each figure legend if possible

g) More discussion of the applications and limitations of this model to future in-vivo dopamine neuron recordings at the single neuron and population levels would be beneficial for potential adopters.

h) Do model parameters differ across different midbrain regions? The in-vitro data may be too limited to address this point, but some discussion may be pertinent.

We look forward to receiving your revised manuscript.

Kind regards,

Gennady Cymbalyuk, Ph.D.

Academic Editor

PLOS ONE

Journal Requirements:

Reviewers' comments:

Reviewer's Responses to Questions

**Comments to the Author**

1. Is the manuscript technically sound, and do the data support the conclusions?

Reviewer #1: Partly

Reviewer #2: Yes

2. Has the statistical analysis been performed appropriately and rigorously? 

Reviewer #1: Yes

Reviewer #2: Yes

3. Have the authors made all data underlying the findings in their manuscript fully available?

Reviewer #1: Yes

Reviewer #2: No

4. Is the manuscript presented in an intelligible fashion and written in standard English?

Reviewer #1: Yes

Reviewer #2: Yes

5. Review Comments to the Author

Reviewer #1: Fleming et al present important empirical and modeling results aimed at inferring spiking from calcium signals in dopaminergic neurons. Although the scope and applicability is somewhat limited, the results will be relevant to the increasing number of labs using calcium imaging approaches to studying in-vivo dynamics in this cell-type.

Although the authors acknowledge that the decay and rate parameters vary across neurons (and therefore the main utility of the model is in interpreting population level dynamics rather than single neurons), it is unclear from the data presented how much the parameters actually vary across neurons and how small changes in the parameter estimates might impact inferred spiking across the population. This point seems critical for determining whether the parameter estimates given can be broadly applied to dopamine neurons in-vivo, especially given the significant variability observed in other regions and cell-types. Specifically:

1. Decay parameter medians and ranges for in-vitro data are shown in Figure 1, but the authors should include data points for each individual neuron, especially given that the number of neurons for each condition is quite low (n’s should be listed in the legend).

2. Each ‘recording’ in Figure 1 was considered an independent measurement even though two recordings were made for some neurons (lines 59-61). It’s not clear whether separate recordings from the same neurons were used to calculate the median decay times used for the in-vivo inference.

3. It would be useful to determine how varying the rate and decay across neurons within some range (ideally estimated from in-vitro recordings), impacts estimates of phasic increases and decreases and slow ramps. Perhaps small changes would not change the population spiking estimates much, but this seems important to know when applying to future in-vivo recordings which may sample from different dopamine neuron populations.

In Figure 4, the authors show that positive and negative ramping spike rates can be recovered from ramping calcium signals in single neurons using their model. However, nothing is shown to validate their inference against electrophysiological data as in Figures 2 and 3. Does their spike inference model reveal features of the ramping that cannot be gleaned from the calcium signal alone (as for reward omission trials)? Other studies (i.e. Wei et al 2020) have argued that negative and positive ramps in single neuron spiking are differently represented in calcium signaling – does this model reconcile those discrepancies? Since the model parameters can vary across single neurons, can the authors show features of spike ramping at the population level that can be accurately recovered from their model (e.g. rates or proportions of positive and negative ramping neurons?)

Other minor points:

1. To account for the rise time kinetics of the indicator, the authors state that they shift the inferred spike times by 4 time steps: some discussion of whether this shift is consistent among the indicators or neurons might be useful, along with a discussion on how this shift might be tuned, and why it is optimal to consider this potential third free parameter in this way rather than build in a double exponential with a rise parameter, as is often used to model these calcium transients

2. The type of GCaMP (m or f) used for imaging experiments should be indicated in each figure legend if possible

3. More discussion of the applications and limitations of this model to future in-vivo dopamine neuron recordings at the single neuron and population levels would be beneficial for potential adopters.

4. Do model parameters differ across different midbrain regions? The in-vitro data may be too limited to address this point, but some discussion may be pertinent.

Reviewer #2: Fleming et al applied a published spike inference algorithm to infer spike firing from calcium imaging data on ventral tegmentum (VTA) dopamine (DA) neurons. The researchers recorded simultaneous in vitro calcium imaging and electrophysiological recording of spike firing from the same VTA neurons. They then calculated decay rate and tuning parameters from the datasets to fit the model for all sample cells. With the decay rate calculated from sample cells and average firing rates from previous in vivo recordings on VTA DA neurons, they further tried to infer spikes from in vivo Ca activities of VTA DA neurons found in earlier studies. They successfully compared their results to those found by independent researchers from in vivo electrophysiological recording in similar tasks. Overall, the findings validate the spike inference algorithm for VTA DA neurons and provides estimated parameters that could be used in future imaging studies to estimate trial-averaged firing, thus making it a potentially useful addition to the literature. However, several technical concerns reduce enthusiasm slightly.

Main Comments

1. The authors do a good job of explaining their parameter estimation procedures from in vitro data, but we note that there is considerable variability in the decay rate, especially for the 6f 37C condition, where they have only 5 cells (9 recordings). That variability presents several issues. First, if the larger variability seen in that case (Fig. 1d) is real, it could be problematic for accurately estimating spiking for VTA DA neurons for this particularly common experimental condition, especially using just a single set of validated parameters (e.g. tau=1 sec, lambda=6 Hz). Second, the variability masks the ability to state whether the systematic trends in Fig. 1d between the different calcium sensors at different temperatures are significant. One would perhaps expect that the body temperature decay per time step is faster (smaller value) than for 30C, and that the 6f sensor would be faster than the 6m sensor at the same temperature. Yet the authors are not able to conclude that given the variability. We thus recommend doubling the number of cells, so it is comparable to that of the other two conditions.

2. Related to the above, we wonder how much error is introduced by the variation in the decay per time step when only the median or mean value of the (small) population is used for all cells. For example, might there actually be two different subpopulations of DA neurons (e.g. tonic and bursty), so that the median/mean does not do well for one or the other? Consistent with that, Fig. 1f (middle) seems to show some particularly large spike estimation errors for the 6f/37C case, even when the firing rate is matched to the electrophysiologically observed rate. The cost of misestimating the decay rate would be helpful to understand, and the authors should generate a similar set of curves as Fig. 1f except with the decay rate along the x-axis.

3. For several neurons/recordings, the estimation tends to predict higher calcium levels than observed in the troughs between bursts – never lower. Can the authors comment on why there is such a systematic error, and perhaps a discussion on what could be done to improve the model (or further caveats on the current model’s limitations)?

4. Although Data is fully available without restrictions, there is no indication of where to obtain the data. According to PLoS requirements, availability upon request is insufficient and some information on where the data can be found (e.g. repository) is recommended.

Other points

a) In Fig. 1f, a tau value of 1 sec is chosen for all the curves. How much improvement is there is the actual fitted tau for each cell is used?

b) Also as a supplement for Fig. 1f, it would be helpful to the reader to illustrate what a difference of 5000 in the van Rossum distance means for the raster plot

c) In the Methods, it is not clearly stated how long after the virus injection surgery the authors waited before performing imaging studies.

d) In the Methods, the authors comment on the need to precisely align the animal on the sphere. The “custom alignment tool” referred to is not particularly transparent. For reproducibility by others, clarifying what this tool is and how it is used would be useful.

6. PLOS authors have the option to publish the peer review history of their article (what does this mean?). If published, this will include your full peer review and any attached files.

Reviewer #1: No

Reviewer #2: No

---

## [Author Response · Author response to Decision Letter 0]

22 Apr 2021

Please see the attached "Reviewer response" document for a full response to the reviewers, including figures.

---

## [Decision Letter · Decision Letter 1]

14 May 2021

Inferring spikes from calcium imaging in dopamine neurons

PONE-D-20-38050R1

Dear Dr. Fleming,

We’re pleased to inform you that your manuscript has been judged scientifically suitable for publication and will be formally accepted for publication once it meets all outstanding technical requirements.

Kind regards,

Gennady S. Cymbalyuk, Ph.D.

Academic Editor

PLOS ONE

Additional Editor Comments (optional):

Reviewers' comments:

Reviewer's Responses to Questions

**Comments to the Author**

1. If the authors have adequately addressed your comments raised in a previous round of review and you feel that this manuscript is now acceptable for publication, you may indicate that here to bypass the “Comments to the Author” section, enter your conflict of interest statement in the “Confidential to Editor” section, and submit your "Accept" recommendation.

Reviewer #1: All comments have been addressed

Reviewer #3: All comments have been addressed

2. Is the manuscript technically sound, and do the data support the conclusions?

Reviewer #1: Yes

Reviewer #3: Yes

3. Has the statistical analysis been performed appropriately and rigorously? 

Reviewer #1: Yes

Reviewer #3: Yes

4. Have the authors made all data underlying the findings in their manuscript fully available?

Reviewer #1: Yes

Reviewer #3: Yes

5. Is the manuscript presented in an intelligible fashion and written in standard English?

Reviewer #1: Yes

Reviewer #3: Yes

6. Review Comments to the Author

Reviewer #1: The authors have done an excellent job of addressing my concerns and the paper is now suitable for publication in my view. This is a well-written and thorough paper that will have high relevance for a growing number of researchers investigating relationships between behavior and cell-type specific neural dynamics (particularly VTA neurons).

Reviewer #3: All the questions from Reviewer1 have been well addressed and the elaborated display of the results to address questions are included in the revised manuscript as main or supplementary figures, which makes the manuscript more solid and easier to interpret by the readers. Thus I have no further issues and agree its publication.

7. PLOS authors have the option to publish the peer review history of their article (what does this mean?). If published, this will include your full peer review and any attached files.

Reviewer #1: No

Reviewer #3: No

---

## [Editor Report · Acceptance letter]

26 May 2021

PONE-D-20-38050R1 

Inferring spikes from calcium imaging in dopamine neurons 

Dear Dr. Fleming:

I'm pleased to inform you that your manuscript has been deemed suitable for publication in PLOS ONE. Congratulations! Your manuscript is now with our production department. 

Kind regards, 

on behalf of

Dr. Gennady S. Cymbalyuk 

Academic Editor

PLOS ONE